# Management of Adrenal Cortical Adenomas: Assessment of Bone Status in Patients with (Non-Functioning) Adrenal Incidentalomas

**DOI:** 10.3390/jcm12134244

**Published:** 2023-06-24

**Authors:** Alexandra-Ioana Trandafir, Mihaela Stanciu, Simona Elena Albu, Vasile Razvan Stoian, Irina Ciofu, Cristian Persu, Claudiu Nistor, Mara Carsote

**Affiliations:** 1Department of Endocrinology, C.I. Parhon National Institute of Endocrinology & Carol Davila Doctoral School, 011863 Bucharest, Romania; alexandratrandafir26@gmail.com; 2Department of Endocrinology, Faculty of Medicine, “Lucian Blaga” University of Sibiu, 550024 Sibiu, Romania; mihaela.stanciu@yahoo.com; 3Department of Obstetrics and Gynaecology, Carol Davila University of Medicine and Pharmacy & University Emergency Hospital, 050474 Bucharest, Romania; 4Department 10—Surgery, General Surgery Department 3, Carol Davila University of Medicine and Pharmacy & University Emergency Hospital, 050474 Bucharest, Romania; alcorstar@gmail.com; 5Department of Obstetrics and Gynaecology, Carol Davila University of Medicine and Pharmacy, 050474 Bucharest, Romania; i.marincas.92@gmail.com; 6Department of Urology, Carol Davila University of Medicine and Pharmacy, 050474 Bucharest, Romania; 7Department 4—Cardio-Thoracic Pathology, Thoracic Surgery II Discipline, Carol Davila University of Medicine and Pharmacy & Thoracic Surgery Department, “Dr. Carol Davila” Central Emergency University Military Hospital, 010825 Bucharest, Romania; ncd58@yahoo.com; 8Department of Endocrinology, Carol Davila University of Medicine and Pharmacy & C.I. Parhon National Institute of Endocrinology, 011863 Bucharest, Romania; carsote_m@hotmail.com

**Keywords:** adrenal, bone, surgery, fracture, osteoporosis, cortisol, DXA, adrenalectomy, incidentaloma, TBS

## Abstract

Our aim is to analyse the bone profile in adults with (non-functioning) adrenal incidentalomas (AIs), specifically addressing the impact of autonomous cortisol secretion (ACS). This narrative review, based on a PubMed search from inception to February 2023 (case reports, non-ACS, and other secondary causes of osteoporosis were excluded), included 40 original studies, a total of 3046 patients with female prevalence (female:male ratio of 1921:1125), aged between 20.5 and 95.5 years old. This three decade-based analysis showed that 37 studies provided dual-energy X-ray absorptiometry (DXA) information; another five studies reports results on bone micro-architecture, including trabecular bone score (TBS), spinal deformity index, and high-resolution peripheral quantitative computed tomography; 20 cohorts included data on bone turnover markers (BTMs), while four longitudinal studies followed subjects between 1 and 10.5 years old (surgical versus non-adrenalectomy arms). Post-dexamethasone suppression test (DST) cortisol was inversely associated with bone mineral density (BMD). TBS predicted incidental vertebral fractures (VFx) regardless of BMD, being associated with post-DST cortisol independently of age and BMD. Low BTMs were identified in ACS, but not all studies agreed. An increased prevalence of ACS-related osteoporosis was confirmed in most studies (highest prevalence of 87.5%), as well as of VFx, including in pre-menopause (42.5%), post-menopause (78.6%), and male patients (72.7%) depending on the study, with a 10-fold increased incidental VFx risk up to a 12-fold increased risk after a 2-year follow-up. No specific medication against osteoporosis is indicated in ACS, but adrenalectomy (according to four studies) should be part of the long-term strategy. This bone profile case sample-based study (to our knowledge, one of the largest of its kind) showed that AIs, including the subgroup designated as having ACS, embraces a large panel of osseous complications. The level of evidence remains far from generous; there are still no homogenous results defining ACS and identifying skeletal involvement, which might be a consequence of different investigation clusters underling adrenal and bone assessments over time. However, bone status evaluations and associated therapy decisions remain an essential element of the management of adults with AIs-ACS.

## 1. Introduction

Adrenal incidentalomas (AIs), clinically silent adrenal masses that are accidentally detected during various imaging procedures, are either unilateral or bilateral (10% are bilateral), either benign (majority) or malign, and either functioning or non-functioning from a hormonal perspective; most of these tumours do not display a clearly clinically evident endocrine activity, although some adenomas of the adrenal cortex (currently called “adrenal cortical adenomas”) demonstrate “autonomous cortisol secretion” (ACS), previously called “subclinical Cushing’s syndrome” (SCS), “subclinical hypercortisolism” (SH), or even “preclinical CS” in older publications [1,2,3].

### 1.1. AI and Potential Cortisol Excess

While the radiological point of view concerns AI as a strictly incidental finding, the endocrine point of view typically relates to a negative hormonal profile and a low expected rate of growth (generally that associated with a cortical adenoma). The overall prevalence of AI is 2%, (between 1% and 8.7% depending on the study criteria) [4,5,6]. AI prevalence increases with age, being very rare in children and adolescents (around 0.2%) and reaching up to 7–10% in patients older than 70 years old (with female predominance) [7,8,9,10].

ACS affects between 5% and 30% of all individuals diagnosed with AI [11,12], subtle cortisol secretion being first described by Beierwaltes in 1974 [13]. The proportion of individuals experiencing cortisol excess in small amounts or intermittent patterns varies due to a heterogeneous diagnostic criteria and cut-off points. Currently, the term ACS is preferred to SCS, but either term involves an associated risk of some comorbidities and alterations of hypothalamic–pituitary–adrenal (HPA) axis regulation due to adrenal cortex-associated hormonal autonomy in the absence of the classical signs or symptoms of overt hypercortisolism (for instance, striae rubrae, proximal muscle weakness, facial plethora, easy bruising, purple striae, etc.) [14]. The standard dynamic test for defining ACS is the 1-mg dexamethasone suppression test (DST), but cut-offs regarding the second-day plasma cortisol vary between 1.8 and 5 µg/dL [15,16,17,18,19]. These values are associated with different risks of metabolic, cardiovascular, and skeletal complications [20].

### 1.2. Cortisol Overproduction Targeting Bone Status

Glucocorticoid excess has a damaging effect on bone mass and quality, being the most common cause of secondary osteoporosis [21]. The condition, underlying increased bone resorption and decreased bone formation, is associated with a reduction in bone formation through suppression of osteoblasts activity mediated by upregulation of peroxisome proliferator-activated receptor (PPAR)-γ and inhibition of the wingless (wnt)/β-catenin signalling pathway [22,23,24]. Additionally, sclerostin is produced by osteocytes and has been recognized as a key negative regulator of bone formation, while chronic glucocorticoid exposure can induce autophagy in osteocytes and consequently decreased sclerostin concentrations [25,26]. Cortisol excess stimulates bone resorption through an alteration of the receptor activator of nuclear factor kappa-Β ligand (RANKL)/osteoprotegerin ratio [27,28]. RANKL is a regulator and activator of osteoclasts, while osteoprogerin acts as a decoy receptor for RANKL, preventing its interaction with RANK and causing the inhibition of osteoblastogenesis [29,30,31]. The severity of hypercortisolism-associated skeletal effects also depends on one’s sensitivity to hormones, as, for instance, has been shown by polymorphism studies of glucocorticoid receptor [32]. In the general population, *BclI* and *N363S* polymorphisms are associated with an increased sensitivity to glucocorticoids and a low bone mineral density (BMD) [33,34], while *ER22/23EK* is correlated with reduced sensitivity to glucocorticoids [35].

### 1.3. ACS-Associated Spectrum

While patients suffering from (overt) CS have a clearly established picture of complications, the morbidity in subjects with ACS is less evident; some studies have suggested a higher risk of arterial hypertension, diabetes mellitus, obesity, dyslipidaemia, and osteoporosis, including complications with vertebral fractures (VFx) and damage to the bone microarchitecture, as reflected by reduced trabecular bone scores (TBS). Currently, dual energy X-ray absorptiometry (DXA) remains the gold standard for bone status assessment [36,37]. Bone turnover markers (BTMs) have been evaluated as part of the cortisol-induced abnormalities; for instance, osteocalcin, a major non-collagenous protein produced by osteoblasts, serving as an indicator of bone formation, is affected since cortisol excess induces apoptosis of osteoblasts, thereby decreasing bone formation [38]. The management of ACS-associated osteoporosis/fractures should be approached based on general strategies varying from anti-resorptives to bone-forming agents [39,40,41,42].

### 1.4. Aim

Our purpose is to analyse the profiles (in terms of osteoporosis prevalence, fracture risk assessment, BTMs, TBS, mineral metabolism, and impact of adrenalectomy on bone status) of patients with non-functioning AI (or AI), specifically addressing the subgroup confirmed with ACS (or SCS).

## 2. Materials and Methods

This narrative review is based on a literature search applying the following inclusion criteria: full-length articles accessed via PubMed (English language); search terms “adrenal incidentaloma” (alternatively, “adrenal cortex adenomas”) and bone status (as reflected by key words such as: “osteoporosis”, “bone”, “fracture”, “skeleton”, “trabecular bone score”, “osteocalcin”, “bone turnover marker”); and timeline: between inception and February 2023. The exclusion criteria were studies specially addressing clinically manifest (overt) CS, Conn syndrome, pheochromocytoma, adrenocortical carcinoma, secondary causes of osteoporosis and fractures, and, as types of papers, reviews, case reports, and case series, nor did we include other types of persistent glucocorticoids excess, as seen in Cushing’s disease, paraneoplasia, and iatrogenic CS.

Overall, we identified 40 original studies on patients diagnosed with AI and/or ACS (or SCS) for whom bone assessments of various types were provided; a total of 3046 patients were included, with a higher prevalence of women (female-to-male ratio of 1921:1125, aged between 20.5 and 95.5 years old) [43,44,45,46,47,48,49,50,51,52,53,54,55,56,57,58,59,60,61,62,63,64,65,66,67,68,69,70,71,72,73,74,75,76,77,78,79,80,81,82] (Figure 1).

## 3. Results

### 3.1. Assessing ACS (or SCS) in Patients with Adrenal Tumours

The studies that addressed bone status in patients with adrenal tumours with apparent non-functioning profiles started from 1992 (the only studies that we included with overt CS were those with a studied subgroup of patients diagnosed with SCS or ACS). We report them below from a timeline perspective with respect to various criteria for defining ACS, as mentioned (Table 1).

The mentioned studies used various endocrine criteria in order to define the cortisol excess as seen in ACS (or equivalent) [43,44,45,46,47,48,49,50,51,52,53,54,55,56,57,58,59,60,61,62,63,64,65,66,67,68,69,70,71,72,73,74,75,76,77,78,79,80,81,82]. (Table 2).

According to the aforementioned original studies, ACS, SCS, SH, or preclinical CS was confirmed based on hormonal tests that varied over time, including DST with different doses and cut offs, sometimes in combination with low ACTH (adrenocorticotropic hormone), high UFC (urinary free cortisol), or increased midnight salivary (or serum) cortisol, in addition to lack of clinical phenotype suggestive for CS [43,44,46,47,49,72,73,75,76,77,78,81]. Another type of analysis enrolled patients with AIs as a general category that sub-included individuals with or without SH, not subjects with AI versus SH (or ACS), as most of the cohorts [65].

Older studies introduced the term “preclinical” CS. For instance, Reincke et al. [43] investigated the prevalence of ACS in 68 patients with AI (female-to-male ratio of 44:24; aged between 25 and 90 years old), and 12% of them had positive hormonal activity consistent with the diagnosis of preclinical CS. Subjects with non-functioning AIs and preclinical CS did not have osteoporosis; only 37.5% of the individuals with overt CS did. Preclinical CS was defined as the following: no clinical signs of CS and lack of suppression with regard to serum cortisol levels (<3 µg/dL) after 1-mg and 8-mg DSTs. The rate and severity of associated arterial hypertension, obesity, and diabetes mellitus markedly improved after adrenalectomy in individuals with preclinical CS (after a mean follow-up of 28 months), in addition to achieving the normal cortisol suppression after DST [43].

The same prevalence of preclinical CS (12%) was reported by Ambrosi et al. [44]. Thirty-two patients with incidentally discovered adrenal tumours (female-to-male ratio of 23:9; aged between 28 and 74 years old) were tested for cortisol levels that were not adequately suppressed (<140 nmol/L) after 1-mg DST and loperamide testing (16 mg). Despite identifying the sub-group with preclinical CS among subjects with AIs, the entire AI group had a statistically significantly reduced BTM profile in terms of osteocalcin, carboxy-terminal cross-linked telopeptide of type I collagen (ICTP), and amino-terminal propeptide of type III procollagen (PIIINP) versus age-matched controls (osteocalcin: 3.9 ± 0.6 versus 5.4 ± 0.15 µg/L; ICTP of 2.4 ± 0.1 versus 4.1 ± 0.1 µg/L; PIIINP of 2.2 ± 0.15 versus 3.3 ± 0.2 µg/L; *p* < 0.01), thus proving that underlying cortisol activity might be involved regardless of the specific assays that are used [44].

### 3.2. BMD Analysis in Patients with ACS

DXA assessment of patients with AI (or non-functioning AI) provided the prevalence of osteoporosis/osteopenia or the BMD reports versus controls, time-dependent BMD changes, or post-adrenalectomy bone effects, including on DXA exams. As mentioned, ACS (or SCS or SH) involves a distinct type of tumour with positive (yet mild, not overt) cortisol (persistent) excess, and consecutive DXA-BMD evaluations depend on the criteria for defining this hormonal activity, on the age and sex of the studied population (including the menopausal status of women), and on the sample size of the cohort. Except for three, 37 cohorts provided data in terms of DXA and/or VFs [43,47,48,49,50,51,52,53,54,55,56,57,58,59,60,61,62,63,64,65,66,67,68,69,70,71,72,73,74,75,76,77,78,79,80,81,82], and a confirmation of a negative impact of bone status was generally confirmed.

The first type of mentioned analysis showed a higher rate of osteoporosis in ACS versus non-functioning AI. For example, Ueland et al. [77] found rates of 18.1% versus 8.5%, respectively (in this study, ACS was defined as serum cortisol greater than 50 nmol/L after 1-mg DST) [77]. Podbregar et al. [79] followed 67 patients with non-functioning AIs (female-to-male ratio of 47:20, mean age of 57.9 years old), and 22% of them progressed to mild ACS (MACS) (diagnosed based on serum cortisol ≥ 50 nmol/L after 1-mg DST), which was reflected by an increased rate of osteoporosis (from 17.9% at baseline to 26.9% after follow-up, *p* = 0.031) [79].

The study of Izawa et al. [82] included 237 adults with adrenal adenomas associated with cortisol excess (CS or MACS), and 47.2% were confirmed with osteoporosis/osteopenia, with greater prevalent in women than men (54.4% versus 23.6%, *p* < 0.001), and 1-mg DST cortisol levels were positively associated with the presence of osteoporosis/osteopenia (OR = 1.124, 95% CI: 1.070–1.181, *p* < 0.001), including in the MACS group (OR = 1.156, 95% CI: 1.046–1.278, *p* = 0.005) [82].

Another cohort enrolled subjects with SH (as defined by at least two positive elements of the following: 1 mg DST—cortisol > 83 nmol/L, UFC > 193 nmoL/24 h, and ACTH < 2.2 pmol/L) who had lower lumbar spine and femoral neck BMD than individuals without SH (*p* = 0.001, respectively, *p* < 0.0001), with associated higher prevalence rates of VFx and osteoporosis (68.8% versus 18.5%, *p* < 0.0001; respectively, 87.5% versus 27.8%, *p* < 0.0001) [66].

Tauchmanova et al. [51] found anomalies of the finger amplitude-dependent speed of sound (Ad-SoS, which has capacity to detect the structural characteristics of bone changes) as measured in SCS and overt CS subgroups and compared them with healthy matched controls (*p* < 0.001, all). BMD and fracture prevalence in SCS were similar to those in overt CS and not to the controls [51].

While most studies have confirmed impairment of bone status in terms of low BMD or a higher prevalence of osteoporosis in ACS, other studies have revealed equivocal results. For example, Rossi et al. [47] showed the absence of significant bone loss in patients with AIs, even associated with SCS (as defined by an abnormal response to at least two standard tests exploring the HPA axis: 2 mg DST—cortisol > 3 µg/dL, UFC > +2 SD of the control group, mean daily cortisol > +2 SD of the control group, and reduced ACTH levels), compared with a sex- and age-matched healthy population [47]. Similarly, Osella et al. [49] revealed no BMD difference between subjects with AI (N = 27) and controls, regardless of the presence of SCS in a small subgroup (as established by the confirmation of two abnormal results with respect to the lack of cortisol suppression after DST (>5 µg/dL), elevated UFC (>216 µg/24 h), low ACTH, and elevated night-to-day cortisol ratio [49].

A study conducted by Ahn et al. [74] in 109 patients with SH [as diagnosed by post-1 mg DST cortisol > 138 nmol/L or >61 nmol/L plus ACTH < 2.2 pmol/L or DHEA-S (dehydroepiandrosterone sulphate) < 2.17 μmol/L in men or <0.95 μmol/L in women] versus 686 subjects with non-functioning AIs found that premenopausal women with SH had significantly lower BMD at lumbar spine (by 9.1%, *p* = 0.008), and femoral neck (by 9.5%, *p* = 0.012) versus premenopausal subjects with non-functioning AIs, while postmenopausal women with SH had statistically significant lower BMD only at the lumbar level (by 7.1%, *p* = 0.016). The prevalence of VFs was similar in premenopausal women (0% versus 1.7%, *p* = 0.578), postmenopausal women (0.0% versus 4.0%, *p* = 0.209), and men (0.0% versus 0.7%, *p* = 0.999). As collateral observations, DHEA-S was positively correlated with lumbar BMD in postmenopausal participants (β = 0.096, *p* = 0.001) and men (β = 0.029, *p* = 0.038). An additional analysis compared the baseline characteristics of the subjects with SH to the Caucasian population (N = 85) from a previous study; overall, the prevalence of osteoporotic fractures was higher in the Caucasian population than the Asian population (*p* < 0.001) [74] (Table 3).

### 3.3. TBS Anomalies Due to ACS-AIs

Mild cortisol overproduction in patients with ACS (or SCS) might impair the skeleton’s qualitative features, as reflected by bone microarchitecture analysis. Overall, we identified five such studies, particularly addressing TBS but also, spinal deformity index (SDI) and high-resolution peripheral quantitative computed tomography (HR-pQCT) [63,72,73,81]. The potential cortisol over-production from the tumour correlates with a negative impact at the level of the micro-architecture, while the level of statistical evidence is less convincing than that seen with DXA-BMD analysis.

Eller-Vainicher et al. [63] assessed TBS in patients diagnosed with AIs (N = 34 subjects with SH and 68 individuals without SH) and 70 controls and revealed that both bone quality and bone mass in SH (as diagnosed by at least two of the following: UFC > 70 µg/24 h, 1 mg DST cortisol > 3 µg/dL, and ACTH < 2.2 pmol/L) were altered: individuals with SH had lower lumbar spine BMD (−0.31 ± 1.17 SD), femoral neck BMD (−0.29 ± 0.91 SD), and TBS (−3.18 ± 1.21) than patients without SH (0.31 ± 1.42 SD, *p* = 0.03; 0.19 ± 0.97 SD, *p* = 0.01; −1.70 ± 1.54, *p* < 0.0001) and controls (0.42 ± 1.52 SD, *p* = 0.02; 0.14 ± 0.76 SD, *p* = 0.02; −1.19 ± 0.99, *p* < 0.0001). TBS was independently associated with incidental VFx and serum cortisol levels. The prevalence of VFx was higher in subjects with SH versus non-SH (82.4% versus 45.6%), while SDI was also increased (1.94 ± 2.24 versus 1.01 ± 1.58) [63].

Cross-sectional analysis of Vinolas et al. [72] proved that TBS is more useful than BMD in individuals associated with various degrees of persistent hypercortisolism (N = 110). Persons with MACS (N = 39, mean age of 57.8 ± 9.3 years) had lower TBS than patients with non-functioning AIs (N = 18, average age of 59.2 ± 9.1 years): 1.30 ± 0.09 versus 1.37 ± 0.12 (*p* < 0.04) but similar BMD, as shown by the values of 1.06 ± 0.20 versus 1.11 ± 0.18 SD (*p* = 0.34). During the mid-term evaluation at 15.5 ± 4.8 months following remission of CS (N = 53), the TBS increase was greater than the BMD increase (10% versus 3%, *p* < 0.02). In patients with overt CS and MACS, no difference was observed regarding TBS and BMD between hypogonadal and eugonadal subjects (1.275 ± 0.10 versus 1.298 ± 0.13, and 1.04 ± 0.18 versus 1.11 ± 0.16 SD, respectively), thus confirming that hypogonadism might not be an essential contributor in such cases [72].

Kim et al. [73] pointed out that lumbar BMD and TBS according to DXA were similar between men with SH (N = 30) and non-functioning AIs (N = 213), while women from the same two subgroups (N = 31 versus N = 142) had statistically significantly lower lumbar BMD (by 6.5%, *p* = 0.026) and TBS (by 2.2%, *p* = 0.04). VFx prevalence was higher in men with CS than in those with AIs (25% versus 1.2%) and those with SH (0.0%, *p* = 0.001), while women experienced a similar prevalence for AIs (3.2%), SH (0.0%), or CS (0.5%, *p* = 0.418). Notably, SH was confirmed by cortisol levels after 1-mg DST of >5.0 μg/dL or >2.2 μg/dL plus ACTH < 10 pg/mL or DHEA-S < 80 μg/dL in men or <35 μg/dL in women [73].

Alternatively, HR-pQCT was used to reflect trabecular bone micro-architectural derangement at the distal radius in one study of 45 subjects with non-functioning AIs (as pointed out by the levels of serum cortisol ≤ 1.8 µg/dL after 1-mg DST) versus 30 individuals with ACS (having cortisol levels between 1.9 and 5.0 µg/dL after 1-mg DST) with a median age of 59 or 60 years old, respectively. Lumbar aBMD was lower in ACS than AIs and was similar at the femoral neck and third distal radius, as were HR-pQC-based trabecular vBMD (*p* = 0.03), inner zone of the trabecular region (*p* = 0.01), bone volume-to-tissue volume ratio (*p* = 0.03), and trabecular thickness (*p* = 0.04). However, the prevalence of osteoporosis was similar between the groups (75% versus 65%, *p* = 0.55), as was the rate of fragility fractures (73.7% versus 55.6%, *p* = 0.24) [75] (Table 4).

### 3.4. BTMs with Regard to AI-ACS

BTMs represent additional tools in skeletal status assessments as a close reflection of persistent hypercortisolism and associated skeleton and mineral metabolism changes. Cortisol overproduction (even mild) might impair bone formation, but overall data are based on small sample-size studies (of fewer than 100 patients per paper) [44,45,46,48,49,50,51,52,53,54,55,56,57,63,65,70,75,76,78,81,82].

Osella et al. [45] showed that patients with AI had a statistically significant osteocalcin reduction and a mild increase in ICTP (marker of bone resorption) compared to controls (6.6 versus 7.8 ng/mL, *p* < 0.05 and 4.2 versus 3.1 µg/L, *p* < 0.01); other BTMs, including bone alkaline phosphatase, did not reach statistical significance, suggesting that osteocalcin is more sensitive than bone alkaline phosphatase in to reflecting the actions of glucocorticoids on bone in AIs [45].

The study by Sartorio et al. [46] included subjects with active CS (N = 12), preclinical CS (N = 6), and AIs (N = 35) and healthy controls (N = 28). As expected, in patients with active CS, osteocalcin (0.9 ± 0.2 ng/mL), ICTP (2.7 ± 0.2 ng/mL), and PIIINP (1.9 ± 0.2 ng/mL) were statistically significant lower than in controls (*p* < 0.0001), but also, in preclinical CS, osteocalcin (2.5 ± 0.8 ng/mL), ICTP (2.2 ± 0.1 ng/mL), and PIIINP (2.2 ± 0.2 ng/mL) were lower than in controls (*p* < 0.0001, *p* < 0.0001 and *p* < 0.02, respectively). In patients with AIs, decreased osteocalcin (4.2 ± 0.5 ng/mL) and ICTP (2.9 ± 0.2 ng/mL) versus controls were identified (*p* < 0.05, respectively, *p* < 0.001) [46].

A small-sample size study conducted by Torlontano et al. [48] confirmed altered osteoblastic activity, as reflected by lower osteocalcin levels, in the subgroup with SH (N = 8) compared to controls (N = 64): 6.8 ± 3.5 versus 8.8 ± 3.2 ng/mL (*p* < 0.005), as was BMD at each central site (*p* < 0.05). PTH was higher (*p* < 0.05) in individuals with SH than in those who were SH-free (N = 24) and in both groups compared to controls (57.1 ± 13.6, 46.0 ± 14.8, and 37.2 ± 10.9 pg/mL, respectively) [48]. Francucci et al. [53] studied a cohort with CS (N = 15), AI (N = 23) and controls (N = 20) and found a significant reduction in lumbar and femoral neck BMD Z-scores (*p* < 0.05) in CS subjects versus AI subjects and controls, as well as in osteocalcin and serum phosphorus levels in CS and AI subjects versus controls (*p* < 0.05) [53].

Hadjidakis et al. [54] analysed DXA and BTMs in menopausal women with AIs (N = 42), demonstrating decreased BMD in patients with SH (N = 18) compared to those without SH (N = 24) at the femoral neck level (BMD of 0.72 ± 0.09 versus 0.78 ± 0.1 g/cm^2^; Z-score of −0.20 ± 0.08 versus 0.43 ± 0.94 SD, *p* < 0.05). Moreover, the frequency of cases with T-scores within the osteopenia range was higher in women with SH (but not for osteoporosis ranges). Serum osteocalcin was lower in women with SH compared to those who were SH-free (18.6 ± 8.6 versus 26.2 ± 8.1 ng/mL, *p* < 0.01), and there were similar PTH values (43 ± 15.6 versus 41.2 ± 14.8 pg/mL, *p* = 0.72); PTH was negatively correlated with femoral neck BMD (r = −0.46, *p* < 0.05) in the SH group (SH defined based on cortisol assay > 70 nmol/L after LDDST) [54].

Another study focused on urinary BTM, namely urinary N-terminal crosslinking telopeptide of type I collagen, as a bone resorption marker. Fifty-five individuals with MACS (mean age of 61.5 ± 10.1 years old) were compared to 12 cases of non-functioning AIs (average age of 66.0 ± 8.9 years old), and there were higher values of this marker (50.6 ± 25.6 versus 26.9 ± 16.6 nmoL BCE/mmolCr, *p* = 0.017) [78].

One of the largest cohorts on BTMs was published by Athimulam et al. [76] in 213 individuals with CS (N = 22), MACS (N = 92), and non-functioning AIs (N = 99). Osteocalcin was increased from one subgroup to another: 14.8 versus 20.1 versus 21.3 ng/mL (*p* = 0.003), as was PINP of 34.8 versus 48.7 versus 48.5 µg/L (*p* = 0.003). C-terminal telopeptide of type I collagen was similar among the three groups (*p* = 0.15). Severity of cortisol excess was found to be inversely correlated with sclerostin measurements, the lowest value being in patients with CS (419 versus 538 versus 624 pg/mL, *p* < 0.0001). Sclerostin might become an additional tool to identify subjects with MACS who are at higher risk for osteopenia or osteoporosis (OR = 0.63; 95% CI: 0.40–0.98 for each 100 pg/mL of sclerostin increase, *p* = 0.04). Patients had osteopenia and osteoporosis in different proportions; for CS, the rates were 62% and 38%, respectively; for MACS, they were 56% and 21%; and for non-functioning AI, they were 52% and 19% [76]. In contrast, another study in patients with SH versus non-SH showed similar femoral neck BMDs, as well as osteocalcin and urinary deoxypyridinoline [50]. Overall, 20 studies provided different results on BTMs and mineral metabolism assays (Table 5).

### 3.5. Bone Assessment in Patients with Unilateral Versus Bilateral AIs

Approximately, one out of 10 cases with AI has a bilateral tumour, but a direct relationship with a more damaged bone profile still represents an unresolved issue, mostly due to the lack of large cohorts specifically referring to bilateral, rather than unilateral, adenomas with respect to skeleton status. We found two studies of bilateral lesions (please see Table 1 and Table 2). Morelli et al. [65] showed a similar SH prevalence (SH was diagnosed in the presence of at least two elements: serum cortisol levels after 1-mg DST of >83 nmol/L, UFC of >193 nmol/24 h; and ACTH levels of <2.2 pmol/L) among subjects confirmed with bilateral AI and unilateral AIs (26.3% versus 23.4 %, *p* = 0.680); bilateral AIs were correlated with lower BMD and a higher prevalence of VFx than unilateral AIs (52.6% versus 29.7%, *p* = 0.005); and the diagnosis of VFx was associated with bilateral AIs after adjusting for SH (OR = 1.77, 95% CI 0.85–3.7, *p* = 0.12). The prevalence of VFx tended to be higher in persons with bilateral AIs and SH than in bilateral AIs without SH (70% versus 46.4%, *p* = 0.07) [65].

Ognjanovic et al. [71] evaluated 152 patients (105 with unilateral AIs and 47 with bilateral AIs) and identified a prevalence of SH higher in bilateral than unilateral AIs (29.8% versus 16.3%, *p* = 0.058) according to the serum cortisol levels after 1 mg DST or after LDDST of >50 nmol/L with at least one of the following parameters: midnight serum cortisol of >208 nmol/L, UFC of >245 nmol/24 h or ACTH of <10 ng/L. Participants with bilateral AIs had lower lumbar BMD when compared to unilateral AIs (*p* = 0.002) and an increased prevalence of osteoporosis (37.1% versus 15.9%, *p* = 0.011) [71].

### 3.6. Prevalent Fractures in Individuals with AI and ACS

Several studies addressed the issue of prevalent VFx (rarely all types of fragility fractures, all referring to non-vertebral) (Table 1 and Table 2). Additional factors, such as hypogonadism (such as menopause), might contribute to VFx, although not all studies agree [52,55,59].

One study revealed that the prevalence of VFx was similar between SCS and overt CS (69% versus 57%, *p* = 0.56), including clinical VFx (28% versus 11.4%, *p* = 0.22) and multiple VFx (36% versus 31%, *p* = 0.92). VFx was independently associated with cortisol-to-DHEAS ratio. Lumbar BMD and cortisol-to-DHEAS ratio were the best predictors of VFx (*p* < 0.01) [56]. Another study reported a VFx prevalence of 35.1% among patients with AIs. The patients with SH experienced a higher rate of VFx (81.5%) after a 2-year follow-up compared to baseline (55.6%, *p* = 0.04) and deterioration of SDI (2.11 ± 1.85 versus 1.11 ± 1.47, *p* = 0.032), indicating the importance of periodic check-ups. Additionally, the incidental VFx rate in subjects with SH was increased versus non-SH (48% versus 13%, *p* = 0.001). The risk of developing new VFx was independently associated with the presence of SH in apparently non-functional adrenal tumours (OR = 12.3, 95% CI 4.1–36.5, *p* = 0.001). In this study, SH was defined by the presence of at least two alterations among cortisol levels after 1-mg DST > 3 µg/dL, increased UFC levels > 70 µg/24 h, and low ACTH levels < 10 pg/mL [62]. Similarly, Lasco et al. [67] showed that SH was associated with a higher prevalence of VFx independently of BMD; while subjects with SH also had reduced lumbar BMD versus non-SH subjects (*p* < 0.01) [67].

Morelli et al. [69] identified that serum cortisol levels after 1-mg DST greater than 2.0 µg/dL offered 73.6% sensitivity and 70.5% specificity for prevalent VFx detection compared to 80% sensitivity, and 68.8% specificity for incident VFx identification [69]. Li et al. [80] published a large cross-sectional study of individuals diagnosed with adrenal adenomas (N = 1004) and age/sex-matched referent subjects (N = 1004), with MACS confirmed in 8% of the subjects and non-functioning AIs in 14% of the cohort with an associated elevated prevalence of any fracture in these two subgroups (47.9% versus 41.3%, *p* = 0.003) and of VFx (6.4% versus 3.6%, *p* = 0.004). During follow-up, patients with adrenal adenomas had a higher cumulative incidence of new fractures compared to controls (OR = 1.27; 95% CI: 1.07–1.52; 3.5% versus 3.6%, *p* = 0.33) [80].

Yano et al. [81] included 194 patients with adrenal tumours with ACS (N = 97) and non-functioning AIs (N = 97); ACS was associated a higher rate concerning the co-presence of VFx and arterial stiffness (23% versus 2%, *p* < 0.001) and of VFx and abdominal aortic calcifications (22% versus 1%, *p* < 0.001) compared to subjects with non-functioning AIs [81].

Notably, menopausal status might influence the rate of VFx. Chiodini et al. [55] reported VFx in half of women with SH (70 patients and 84 controls). SH was associated with a higher prevalence of fractures in patients with SH than in controls without SH; in premenopausal women, the prevalence rates were 42.9% versus 0% (*p* = 0.001), respectively, compared to 7.1% in non-SH (*p* = 0.049) and in postmenopausal women (SH of 78.6%, controls of 37.7%, *p* = 0.006; non-SH of 42.9%, *p* = 0.024). A BMD-based Z-score significantly predicted the prevalence of fractures in AI patients (OR = 0.50; 95% CI, 0.31–0.80; *p* = 0.004) [55]. Another study of 287 subjects with AIs (non-SH: N = 202, and with SH: N = 85) and controls (N = 194) showed a higher VFx prevalence in patients with SH than non-SH subjects and controls (70.6%, 21.8%, and 22.2%, respectively, *p* < 0.0001) [58].

### 3.7. Longitudinal Studies following Bone Status with Regard to ACS and AIs

The optimal management of patients with ACS has been associated with a dynamic approach over the years due to continuous changes in definition criteria and specific indications, which vary from an individual approach to guideline recommendations. The importance of the topic is related to close follow-up of medical treatment for associated morbidities being needed, especially if surgical treatment is not chosen. Four studies followed patients from 1 year to more than a decade [57,62,79,80] (Table 6).

Regarding an interventional approach specifically addressing medication for osteoporosis in ACS, a limited number of publications were identified. Tauchmanova et al. [57] evaluated the effects of 1-year clodronate (100 mg every week) on lumbar and femoral neck BMD and BTMs in premenopausal women with SCS and osteopenia/osteoporosis (N = 46). Patients were randomized to receive clodronate plus calcium (500 mg daily) and vitamin D3 (800 mg daily) supplements or only calcium plus vitamin D. As expected, clodronate administration increased lumbar BMD (*p* = 0.04) and conserved BMD values at the femoral neck, while incidental fractures occurred only in the non-clodronate group [57]. Currently, the drug is no longer used or available in daily practice.

Alternatively, removal of the tumour might help the bone status, but not all studies agree. A prospective, randomized study of small size compared laparoscopic adrenalectomy (N = 23) to conservative management (N = 22) in SCS (defined by serum cortisol higher than 2 µg/dL after 1-mg DST). Adrenalectomy improved the cardio-metabolic profile underlying hypertension and diabetes mellitus but not bone parameters [60]. A similar result was reported by Iacobone et al. [64], concluding that adrenalectomy is better than medical therapy in improving the values of high blood pressure hypertension but not the values of DXA-based T-scores [64].

A more complex decision involving surgery is needed in cases with bilateral tumours. Perogamvros et al. [68] published a retrospective study of 33 patients with bilateral AIs; the surgical group included 14 patients who underwent unilateral adrenalectomy of the largest adenoma (all women, mean age of 54.9 ± 6.7 years old), and the non-surgical group enrolled 19 patients (14/19 women, average age of 59.0 ± 8.7 years old). An improvement of comorbidities was registered only in the first group, while two of the three subjects with osteoporosis were associated with an increase in post-operative BMD (*p* = 0.03) [68]. Salcuni et al. [70] identified a reduced probability of incidental VFx in patients with SH who underwent adrenalectomy versus those who were surgery-free (N = 55 individuals with SH, of whom 32 people underwent adrenalectomy, and 23 received a conservative approach). Adrenalectomy in subjects with SH was associated with a 30% VFx risk reduction (OR = 0.7, 95% CI 0.01–0.05, *p* = 0.008) regardless of age, gender, follow-up duration, degree of hypercortisolism, lumbar BMD, or prevalent VFx at baseline. In the surgical group, only 9.4% of subjects experienced a new VFx, which was statistically significantly lower than in the non-adrenalectomy group (52.2%, *p* < 0.0001) [70].

Overall, four studies specifically addressed the outcome of adrenalectomy in patients with AIs-ACs (between 2009 and 2016). While the total number of patients per study remains low (33, 35, 45, and 55), all had a control arm (surgery versus conservative approach); the period of follow-up varied from a mean of 54 ± 34 and 53.9 ± 21.3 months in two cohorts (after adrenalectomy), respectively, and a mean of 56 ± 37 or 51.8 ± 20.1 months to a median of 7.7 (2–17) years in one study (non-surgical group) (Table 7).

## 4. Discussions

### 4.1. Prevalence of ACS in Patients with Osteoporosis and Fracture

Additionally, two studies were added from the opposite perspective: the assessment of adrenal (cortisol) status in patients diagnosed with osteoporosis and fractures (N = 320, female-to-male ratio of 275:45, mean age of 62.17 years old). This analysis was distinct from the previously described studies. We examined ACS profiles in patients already diagnosed with osteoporosis and with associated fragility fractures, and we identified two additional (transversal) studies (published in 2021 and 2007) to add to the initial panel of 40 original papers. The hypothesis of a negative effect of ACS on skeletal status was supported by Chiodini et al. [83], showing that SH is more common in adults with osteoporosis than in those with abnormal bone profiles. They found a prevalence of 4.8% for SH (between 1.32% and 8.20%) among patients with osteoporosis, while no subjects without osteoporosis had SH. The authors included 219 participants without clinically overt CS or other obvious secondary causes of bone loss (200 women and 19 men). SH was diagnosed according to incomplete cortisol suppression (>50.0 nmol/L) after LDDST, UFC of >165.6 nmol/L (normal ranges between 22.2 and 165.6 nmol/L), and/or midnight cortisol level of >207 nmol/L (normal ranges between 0.0 and 138.5 nmol/L). Patients with osteoporosis had lower T-scores (BMD) at the lumbar spine of −2.88 ± 1.09 SD versus 1.36 ± 1.09 SD, at the total neck of −1.96 ± 0.88 SD versus −0.99 ± 0.97 SD, and at the femoral neck of −2.02 ± 0.87 SD versus −1.12 ± 0.91 SD compared to individuals without osteoporosis (*p* = 0.001) [83].

The other cohort enrolled 101 subjects with prevalent fragility fractures (75 women and 26 men) with a mean age of 65 ± 10.3 years old. Five of 101 (3 women and 2 men) were diagnosed with “less severe hypercortisolism” (a prevalence of 5%), which was established by unsuppressed serum cortisol less than1.8 μg/dL after LDDST. Lumbar and femoral neck BMDs (Z-score) were similar in individuals with or without this level of cortisol excess as described by Pugliese et al. [84] (Table 8).

### 4.2. Controversies in Defining Subtle (Non-Overt) Cortisol Excess Due to Adrenal Cortex Adenomas

We believe that the most difficult aspect of studying bone profiles in AIs/ACS is the definition of the entities themselves (ACS or SCS) since various criteria and terms have been used according to our 30-year analysis. Pre-CS or preclinical CS historically referred to asymptomatic, but biochemically active hypercortisolism [85,86]. Today, the shift from using the term “SCS” to “ACS” has occurred in daily practice; the key assay remains the serum cortisol level after 1-mg DST [16,17,87,88]. In addition, a value of cortisol between 1.8 µg/dL and 5 µg/dL, low morning plasma ACTH less than 10 pg/mL, and/or increased UFC levels improve the accuracy of ACS diagnosis [89]. Morelli et al. reported that the presence of combined criteria characterized by at least two biochemical anomalies among the values after 1-mg DST cortisol level greater than 3 µg/dL (83 nmol/L), high UFC, and ACTH less than 10 pg/mL (“combined DST-UFC-ACTH criterion”) was associate with sensitivity of 61.9% and specificity of 77.1% and thus accuracy of 75.8% in predicting complications [61]. Similarly, a cortisol value higher than 2 µg/dL (55 nmol/L) after 1-mg DST predicts VFx in patients with AIs with sensitivity of 80% and specificity of 68.8% [69]. In most studies, a combination of at least two abnormal HPA tests, especially DST and another assay (such as baseline assays of UFC or ACTH), were used [46,47,49,50,52,55,57,58,59,61,62,63,64,65,66,68,70,71]. Moreover, two studies included DHEAS assessments of less than 2.17 μmol/L in men and less than 0.95 μmol/L in women or alternatively ACTH values less than the cut off of 2.2 pmol/L in association with 1-mg DST to confirm ACS [73,74].

Overall, the prevalence of ACS (SCS) among patients with AIs varied from 6% to 12% [43,44], being more frequent among subjects with bilateral versus unilateral adenomas, as pointed out, for instance, by the aforementioned cohorts of 41.5% versus 12.2% [90], 35.1% versus 17.9% [91], 21.7% versus 6.2% [92], and 29.8% versus 16.3% [71]. Nevertheless, a similar proportion of SH among individuals with bilateral and unilateral AIs was found by some authors (26.3% versus 23.4%) despite being associated with lower BMD and a higher prevalence of VFx in bilateral tumours compared to unilateral lesions [65].

### 4.3. Bone Status and ACS

The difference in hormonal diagnosis (not only in terminology) represents one major contributor to the heterogeneous bone profiles that we found, in addition to other bias elements, such as different ages (we only identified adult studies), comorbidities (such as menopausal status and, generally hypogonadism, and diabetes mellitus), previous or concurrent medication for osteoporosis, and fracture risk reduction; further, with respect to the tumour itself, bilateral lesions are more prone than unilateral AIs to developing bone profile-associated abnormalities [43,44,45,46,47,48,49,50,51,52,53,54,55,56,57,58,59,60,61,62,63,64,65,66,67,68,69,70,71,72,73,74,75,76,77,78,79,80,81,82] (Figure 2).

Moreover, the heterogeneity of the mineral metabolism and fracture risk evaluations that we found among the 40 original studies should have limited the data of a systematic review, which is why we included a broader spectrum under the umbrella of a narrative review. Based on our research, this three decade-based analysis of published sample case studies enrolled more than 3000 patients with confirmed AIs/ACS; among the 40 studies, 37 provided DXA-based information, five reported results on bone micro-architecture, including TBS, SDI, and HR-pQCT, and 20 cohorts included data regarding BTMs, while four longitudinal studies (surgical versus non-surgical arms) followed subjects between 1 and 10.5 years old [43,44,45,46,47,48,49,50,51,52,53,54,55,56,57,58,59,60,61,62,63,64,65,66,67,68,69,70,71,72,73,74,75,76,77,78,79,80,81,82] (Figure 3).

Bone micro-architecture is affected by chronic cortisol excess, as directly shown by histomorphometry/bone biopsy-based studies [93]. However, SDI, but mostly TBS (during last decade), is applicable in daily practice [94]. As mentioned, four TBS-based studies proved its value as a complementary, easy-to-access tool in adults with AIs/ACS [63,72,73,81]. TBS predicts incidental VFx regardless of BMD, being associated with post-DST cortisol levels independently of the patients’ age and DXA-BMD [63]. Of course, there are still open questions regarding these subjects, especially in menopausal women diagnosed with type 2 diabetes mellitus, including the exact contribution to TBS damage due to associated glucose profile anomalies. Moreover, SDI is a semi-quantitative method that integrates the number and severity of VFx [95]; an increased SDI has been found in subjects with SH [58,62,63].

Decreased bone formation, as revealed by low osteocalcin, was found in some of the aforementioned studies; of course, this condition is also an effect of the co-presence of diabetes mellitus in certain subgroups [44,45,46,48,53,54,56,76]. Other cohorts did not have mild cortisol overproduction associated with serum osteocalcin anomalies [50,52,78]. Less valuable data are provided for bone alkaline phosphatase, also a bone formation marker, which can be decreased [96] or similar in AIs/ACS and controls [45,48,50]. Additionally, hypercortisolism induces bone resorption, leading to a decrease in sclerostin (secreted by osteocytes). One study found that the severity of cortisol excess was inversely correlated with its levels [76]. Serum C-telopeptide of type I collagen ICTP assays had various results: they were increased [45,56], reduced [44,46], or similar in patients with ACS versus non-ACS [48,76]. Urinary deoxypyridinoline does not seem relevant in ACS/SCS [48,50,52,53]. Urinary N-terminal crosslinking telopeptide of type I collagen, another bone resorption marker, was statistically significant higher in MACS than non-functioning AIs according to one study [78].

Persistent hypercortisolism impairs intestinal calcium absorption and renal tubular calcium reabsorption, potentially leading to secondary hyperparathyroidism [96,97,98,99,100,101]. Some researchers have found a higher level of PTH in subjects with ACS versus non-functioning AIs [48,50], although not all authors [52,54,56]. One study confirmed a negative correlation between serum PTH and lumbar spine and femoral neck BMD [49]. The levels of 25-hydroxyvitamin D are not necessarily discriminative among ACS/non-ACS individuals [63,65,70,75,78].

Moreover, the relationship between anomalies of DHEA-S in AIs/ACS and bone status is still inconclusive [77,102,103]. One study identified that up to 40% of subjects had AI-associate reduced DHEA-S levels [44], while DHEA-S was positively correlated with lumbar BMD [56], but not all studies showed statistically significant data [77].

Overall, the prevalence of osteoporosis complicated or not with fragility fractures, especially at the vertebral level, is higher in individuals with SH versus non-SH, for example, 87.5% versus 27.8%, 68.8% versus 18.5% [66], 75% versus 65%, and 74% versus 55.6% [75]. Others found a similar prevalence in ACS and non-functioning AIs [74,78]. A greater number of VFx than controls was observed in premenopausal women, also consistent with cortisol rather than oestrogens status (VFx prevalence of 42.9% in pre-menopause and of 78.6% in post-menopause) [55]. Another study suggested that subjects with SH had a 10-fold increased risk of new VFx, independent of age, gender, and BMD [69], as well as a 12-fold increase after a 2-year follow-up with a VFx prevalence of 81.5% [62]. A similar rate of 72.7% was reported in adult men (most VFx being asymptomatic), not only in women [59]. We identified only one study applying FRAX in patients older than 40 years old, and its values were similar between subjects with ACS and those with non-functional AIs [75].

### 4.4. Interventional Considerations

The therapeutic approach for patients diagnosed with ACS/SCS is based on non-randomized trials. Weekly clodronate might prevent bone loss and VFx in women with SH [57], but current applications are limited. Adequate calcium and vitamin D are required, while the best treatment strategy against osteoporosis is supported by the general recommendations for adult (primary and/or glucocorticoid) osteoporosis.

Notably, adrenalectomy should be considered when deciding on an anti-osteoporosis strategy (including in cases of unilateral adrenal removal for patients with bilateral AIs) since improvement of bone status is expected after surgery [68,70]; however, some authors showed no remarkable post-operative benefits regarding BMD improvement or fracture prevention [60,64]. Transient post-adrenalectomy adrenal insufficiency, an indirect piece of evidence for prior tumour-related cortisol hypersecretion, might highlight further bone advantages [43,44,104,105].

The precise timing of periodic check-ups (biochemical and imaging evaluation) is still a matter of debate [8,14,16,17,18,19,106,107]. Longitudinal studies have reported an increased frequency of complications, including of osteoporotic fractures, in patients with AIs/ACS [62,79,80]. Additionally, the management (short- and long-term medical and surgical strategies) needs to integrate bone status into the larger panel of cardiovascular and metabolic complications due to ACS/SCS [108,109,110], as well as generally in patients with AIs/non-functioning AIs [111,112,113,114,115,116,117]. “Non-functional” does not necessarily mean harmless, and insulin resistance, obesity, high blood pressure, diabetes, and dyslipidaemia are identified in the same patients who display a higher risk of osteoporosis and VFs [118,119,120,121,122,123,124], as is increased arterial intima-media thickness-associated endothelial dysfunction [125,126], elevated uric acid [127], and pro-coagulation status [128,129], as well as potential elevated all-cause mortality [130].

Cortisol excess due to adrenal tumour-related overproduction might cause glucocorticoid osteoporosis. In patients with ACS/SH, primary (menopausal or age-related osteoporosis) might overlap with type 2 diabetes mellitus-associated bone damage. Despite not having an overt clinical picture of CS, hypercortisolaemia might impair bone status through well-known pathogenic mechanisms of glucocorticoid osteoporosis. The degree of bone influence is related to the duration and level of hormone overproduction, as similarly seen in iatrogenic CS. Glucocorticoids, while acting on osteoblasts, reduce their proliferation and differentiation, also causing elevated apoptosis of osteoblasts and osteocytes and thus reduced bone formation (as mentioned, not all studies identified reduced levels of osteocalcin and bone alkaline phosphatase in subjects diagnosed with ACS/SH). Moreover, cortisol excess induces increased osteoclastogenesis with a greater number of osteoclasts being recruited in association with reduced apoptosis, which is prone to bone resorption. Finally, hormone overproduction impairs the overall bone turnover, favouring a reduced BMD (as pointed out above in subjects with ACS as well). A good level of evidence shows that anti-resorptive medications, such as bisphosphonates and denosumab, as well as bone-forming agents, such as teriparatide, counteract these pathogenic elements with good clinical results; thus, subjects with ACS/SH should be no exception according to the level of evidence that we have so far [131,132,133,134,135].

### 4.5. Further Considerations

Overall, the topic of recognizing mild cortisol excess in these adrenal tumours is mandatory to address the outcome, to decide whether and when surgery is needed, or to select candidates for a specific medication against osteoporosis. As mentioned, the limits of the current topic derive from the lack of standardization both in defining the subgroups with non-overt cortisol excess among patients with apparently non-functional adrenal cortex adenomas and in recommending a standard intervention. We acknowledge that our current work represents a narrative (not a systematic) review, and some studies might be biased because only PubMed was used for the literature search.

Anomalies of bone profiles (including the diagnosis of osteoporosis and fragility fractures) represent an important element in overall decision making; thus, we need not only cross-sectional but also longitudinal data. Future perspectives should address three main issues: the panel of bone-related investigations and its timing during lifelong surveillance to assess the fracture risk; randomized trials concerning medication against osteoporosis and fracture risk reduction in this specific group of patients; and a focus on the role of bone profiles in protocols for deciding on adrenalectomy (including in individuals with bilateral AIs).

## 5. Conclusions

According to our bone profile study of published data that is, to our knowledge, one of the largest on its kind, tumours considered AIs or non-functioning AIs, including the subgroup designated as having ACS or SCS (or SH), includes a large panel of osseous complications from lower BMD and TBS to blunt BTMs and a higher prevalence of osteoporosis and/or VFs compared to non-AIs or non-SCS. The level of evidence remains far from generous; there are still no homogenous results, which might be a consequence of different investigation clusters with respect to adrenal and bone evaluations that have been used over time. However, awareness is mandatory, and bone assessments should be an essential element of the management in these adults.

## Figures and Tables

**Figure 1 jcm-12-04244-f001:**
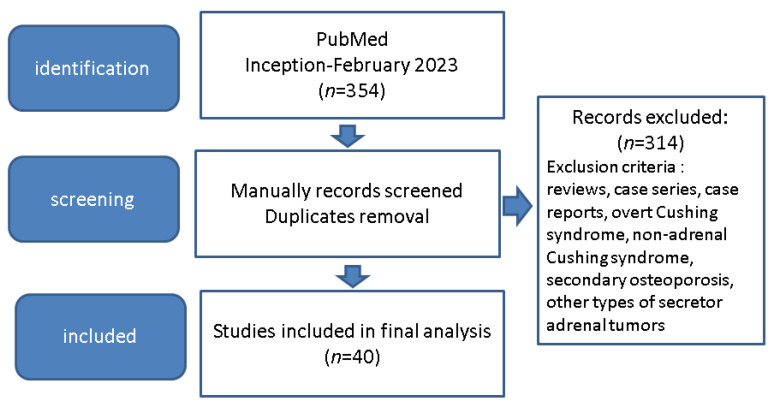
Flowchart diagram of included papers regarding patients with adrenal tumours of AI (non-functioning AI or adrenal cortex adenomas) type (and/or ACS or SCS) in whom bone status was analysed according to our methodology [43,44,45,46,47,48,49,50,51,52,53,54,55,56,57,58,59,60,61,62,63,64,65,66,67,68,69,70,71,72,73,74,75,76,77,78,79,80,81,82].

**Figure 2 jcm-12-04244-f002:**
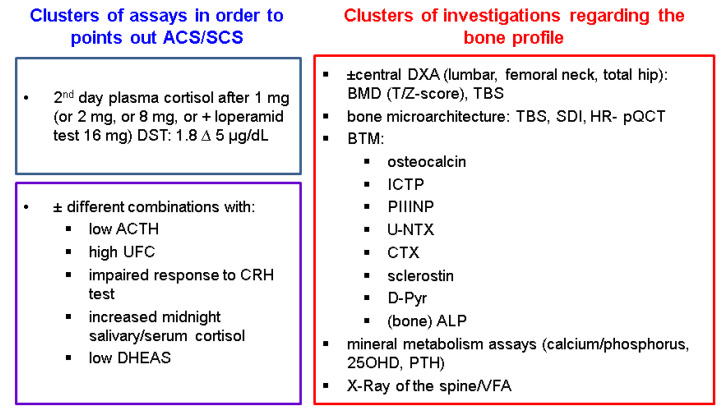
Adrenal and bone assays among mentioned studies [43,44,45,46,47,48,49,50,51,52,53,54,55,56,57,58,59,60,61,62,63,64,65,66,67,68,69,70,71,72,73,74,75,76,77,78,79,80,81,82]. Abbreviations: ACS = autonomous cortisol secretion; ACTH = adrenocorticotropic hormone; BMD = bone mineral density; BTM = bone turnover markers; DXA = dual-energy X-ray absorptiometry; DST = dexamethasone suppression test; ALP = alkaline phosphatase; CRH = corticotropin-releasing hormone; CTX = C-terminal telopeptide of type 1 collagen; DHEA-S = dehydroepiandrosterone sulphate; D-Pyr = urinary deoxypyridinoline; ICTP = carboxy-terminal cross-linked telopeptide of type I collagen; HR-pQCT = high-resolution peripheral quantitative computed tomography; PIIINP = amino-terminal propeptide of type III procollagen; U-NTX = urinary N-terminal crosslinking telopeptide of type I collagen; SDI = spinal deformity index; VFA = vertebral fracture assessment; TBS = Trabecular Bone Score; VFA = vertebral fracture assessment; UFC = urinary free cortisol; SCS = subclinical Cushing’s syndrome.

**Figure 3 jcm-12-04244-f003:**
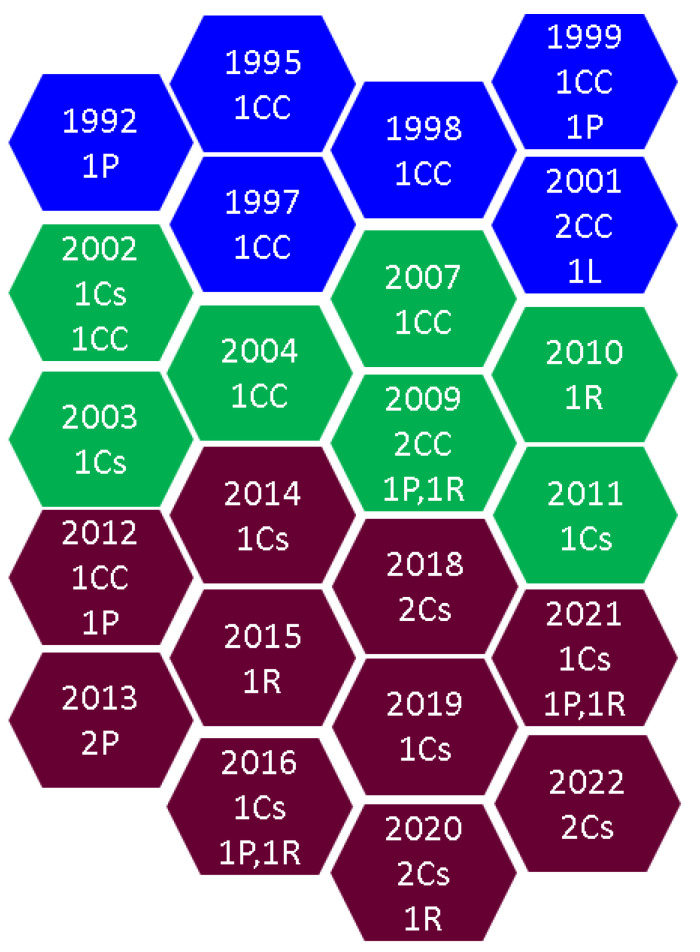
Three-decade hexagon infographic timeline according to our methodology (blue: studies between 1992 and 2001; green: studies between 2002 and 2011; purple: studies between 2012 and 2022) [43,44,45,46,47,48,49,50,51,52,53,54,55,56,57,58,59,60,61,62,63,64,65,66,67,68,69,70,71,72,73,74,75,76,77,78,79,80,81,82]. Abbreviations: CC = case-control study; Cs = cross-sectional study; P = prospective study; R = retrospective study.

**Table 1 jcm-12-04244-t001:** Studies with AI/non-functioning AI and bone assessments: the analysis of criteria defining ACS (or SCS)—the studies are displayed in the order of publication [43,44,45,46,47,48,49,50,51,52,53,54,55,56,57,58,59,60,61,62,63,64,65,66,67,68,69,70,71,72,73,74,75,76,77,78,79,80,81,82]. Studied population [43,44,45,46,47,48,49,50,51,52,53,54,55,56,57,58,59,60,61,62,63,64,65,66,67,68,69,70,71,72,73,74,75,76,77,78,79,80,81,82].

First Author Year of Publication Reference Number Study Design	Studied Population (and Included Subgroups According to Each Study) Gender (F/M) Age (Years)			
Reincke 1992 [43] Prospective study	N = 68 N1 = 8 with overt CS F/M = 8/0 Mean age: 39 ± 8 y (26–50 y)	N2 = 8 with preclinical CS F/M = 5/3 Mean age: 50 ± 15 y (25–71 y)	N3 = 58 with NFAI F/M = 37/21 Mean age: 59 ± 12 y (25–90 y)	
Ambrosi 1995 [44] Case-control study	N = 32 N1 = 32 with AI F/M = 23/9 Median age (women):49 y/median age (men): 67 y	N2 = 4 with preclinical CS F/M = 1/3 (42–72 y)	N3 = 14 controls F/M = 8/6 (25–50 y)	
Osella 1997 [45] Case-control study	N = 22 N1 = 18 with CS F/M = 15/3 Median age: 36 y (15–64 y)	N2 = 22 with AI F/M = 13/9 Median age: 54.5 y (25–73 y)	N3 = 22 controls Median age: 55 y (25–73 y)	
Sartorio 1998 [46] Case-control study	N = 53 N1 = 12 with CS F/M = 10/2 (21–46 y)	N2 = 6 with preclinical CS F/M = 2/4 (42–74 y )	N3 = 35 with AI F/M = 25/10 (34–73 y)	N4 = 28 controls (25–69 y)
Rossi 1999 [47] Prospective study	N = 50 N1 = 50 with AI F/M = 29/21 mean age: 56.3 y (32–73 y) of these 12 with SCS = N2: F/M = 9/3 Mean age: 60.7 y (47–72 y)		N3 = 107 controls F/M = 65/42 Mean age: 52.2 y	
Torlontano 1999 [48] Case-control study	N = 32 women N1 = 8 with SH F/M = 8/0 Mean age: 54.0 ± 18.4 y (26–75 y)	N2 = 24 without SH F/M = 24/0 Mean age: 57.3 ± 10.0 y (38–76 y)	N3 = 64 controls F/M = 64/0 Mean age: 56.4 ± 12.3 y (22–76 y)	
Osella 2001 [49] Case-control study	N = 27 N1 = 27 with AI F/M = 18/9 Median age: 57 y (42–73 y) of these N2 = 8 with SCS F/M = 5/3 Median age: 55.87 y		N3 = 54 controls F/M = 36/18 Median age: 56.5 y (41–73 y)	
Chiodini 2001 [50] Longitudinal study	N = 24 N1 = 7 with SH F/M = 7/0 Mean age: 42.4 ± 16.7 y (26–72 y)	N2 = 17 without SH F/M = 17/0 Mean age: 56.0 ± 11.8 y (38–77 y)		
Tauchmanova 2001 [51] Case-control study	N = 34 N1 =15 with overt CS F/M = 9/6 Mean age: 41.73 ± 10.2 y (21–50 y)	N2 = 19 with SCS F/M = 11/8 Mean age: 44.5 ± 9.8 y (25–59 y)	N3 = 76 controls F/M = 46/30 Mean age: 46.5 ± 13.7 y	
Chiodini 2002 [52] Cross-sectional study	N = 38 N1 = 13 with SH F/M = 0/13 Mean age: 61.4 ± 11.3 y (40–75 y)	N2 = 25 without SH F/M = 0/25 Mean age: 55.2 ± 12.6 y (21–78 y)	N3 = 38 controls Mean age: 56.9 ± 11.8 y (26–74 y)	
Francucci 2002 [53] Case-control study	N = 38 women N1 = 15 with CS F/M = 15/0 Mean age: 46.5 ± 14.8 y	N2 = 23 with AI F/M = 23/0 Mean age: 57.6 ± 10.9 y	N3 = 20 controls F/M = 20/0 Mean age: 51.5 ± 9 y	
Hadjidakis 2003 [54] Cross-sectional study	N = 42 N1 = 18 with SH F/M = 24/0 Mean age: 59.1 ± 6.3 y (48–74 y)	N2 = 24 without SH F/M = 18/0 Mean age: 63.6 ± 5.1 y (53–72 y)		
Chiodini 2004 [55] Case-control study	N = 70 women Premenopausal women: N1 = 7 with SH F/M = 7/0 Mean age: 42.9 ± 3.5 y (26–50 y) N2 = 14 without SH F/M = 14/0 Mean age: 39.3 ± 2.2 y (24–52 y)	N3 = 23 controls F/M = 23/0 Mean age: 41.7 ± 2.0 y (21–54 y) N3 = 61 controls F/M = 61/0 Mean age: 60.8 ± 0.9 y (44–75 y)	Postmenopausal women: N1 = 14 with SH F/M = 14/0 Mean age: 63.9 ± 2.2 y (46–79 y)	N2 = 35 without SH F/M = 35/0 Mean age: 61.5 ± 1.4 y (48–80 y)
Tauchmanova 2007 [56] Case-control study	N = 71 women N1 = 36 with overt CS F/M = 36/0 Median age: 42 y (28–66 y)	N2 = 35 with SCS F/M = 35/0 Median age: 46 y (30–68 y)	N3 = 71 controls Median age: 44 y (28–68 y)	
Tauchmanova 2009 [57] Case-control study	N = 46 N1 = 23 with SCS treated with clodronate F/M = 23/0 Mean age: 42.4 ± 6.4 y	N2 = 23 with SCS (untreated) F/M = 23/0 Mean age: 43.5 ± 6.1 y		
Chiodini 2009 [58] Retrospective study	N = 287 N1 = 85 with SH F/M = 53/32 Mean age: 62.9 ± 9.9 y (34–79 y)	N2 = 202 without SH F/M = 123/79 Mean age: 61.2 ± 11.4 y (21–81 y)	N3 = 194 controls F/M = 104/90 Mean age: 61.1 ± 13.7 y (21–79 y)	
Chiodini 2009 [59] Case-control study	N = 88 men N1 = 22 with SH F/M = 0/22 Mean age: 65.8 ± 10.7 y (42–86 y)	N2 = 66 without SH F/M = 0/66 Mean age: 60.9 ± 12.9 y (21–83 y)	N3 = 90 controls F/M = 0/90 Mean age: 61.8 ± 14.2 (23–90 y)	
Toniato 2009 [60] Prospective study	N = 45 N1 = 23 with adrenalectomy: F/M = 11/12 Mean age: 63 ± 4.1 y	N2 = 22 with conservative management: F/M = 12/10 Mean age: 64 ± 1.8 y		
Morelli 2010 [61] Retrospective study	N = 231 with AI F/M = 120/111 Mean age: 62.9 ± 10.1 y			
Morelli 2011 [62] Cross-sectional study	N = 103 female At baseline: N1 = 27 with SH F/M = 27/0 Mean age: 65.0 ± 8.7 y (41–83 y) N2 = 76 without SH F/M = 76/0 Mean age: 62.7 ± 10.3 y (28–80 y)	After 24 months of follow-up: N1 = 27 with SH F/M = 27/0 Mean age: 67.0 ± 8.5 y (43–85 y) N2 = 76 without SH F/M = 76/0 Mean age: 64.9 ± 10.2 y (30–82 y)		
Eller-Vainicher 2012 [63] Case-control study	N = 102 N1 = 34 with SH F/M = 19/15 Mean age: 66.3 ± 8.3 y (45–79 y)	N2 = 68 without SH F/M = 44/24 Mean age: 67.5 ± 8.7 y (47–83 y)	N3 = 70 controls F/M = 48/22 Mean age: 67.7 ± 14.4 y (37–85 y)	
Iacobone 2012 [64] Prospective study	N = 35 N1 = 22 with adrenalectomy: F/M = 8/12 Median: 55 y (36–78 y)	N2 = 15 with conservative management: F/M = 7/8 Median: 58 y (39–75 y)		
Morelli 2013 [65] Prospective study	N = 213 BAI: N1 = 10 with SH F/M = 4/6 Mean age: 65.6 ± 5.3 y (57–74 y)	N2 = 28 without SH F/M = 20/8 Mean age: 63.7 ± 9.9 y (41–82 y)	UAI: N1 = 41 with SH F/M = 28/13 Mean age: 62.1 ± 11.6 y (39–83 y)	N2 = 134 without SH F/M = 88/46 Mean age: 63.7 ± 11.0 y (31–85 y)
Palmieri 2013 [66] Cross-sectional, prospective study	N = 70 N1 = 16 with SH F/M = 7/9 Mean age: 62.5 ± 10.6 y (34–77 y)	N2 = 54 without SH F/M = 34/20 Mean age: 61.5 ± 10.3 y (39–77 y)		
Lasco 2014 [67] Cross-sectional study	N = 50 women N1 = 3 with SH Median age: 57 ± 3 y	N2 = 47 without SH Median age: 58 ± 4 y		
Perogamvros 2015 [68] Retrospective study	N = 33 N1 = 14 with adrenalectomy: F/M = 14/0 Mean age: 54.9 ± 6.7 y	N2 = 19 with conservative management: F/M = 14/5 Mean age: 59 ± 8.7 y		
Morelli 2016 [69] Retrospective (cross-sectional arm) and prospective (longitudinal arm)	N = 444, of these 126 with AI N1 = 96 patients without incident VFx F/M = 64/32 Mean age: 62.9 ± 9.5 y (27–80 y)	N2 = 30 patients with incident VFx F/M = 16/14 Mean age: 65.5 ± 9.2 y (41–83 y)		
Salcuni 2016 [70] Cross-sectional and prospective interventional study	N = 55 Surgical intervention: N1 = 32 Baseline: F/M = 22/10 Mean age: 61.3 ± 8.1 y (38–75 y) End of follow-up Mean age: 64.7 ± 8.7 y (40–79 y)	Conservative management: N2 = 23 Baseline: F/M = 10/13 Mean age: 65.4 ± 7.05 y (51–75 y) End of follow-up: Mean age: 67.7 ± 6.9 y (53–78 y)		
Ognjanovic 2016 [71] Cross-sectional study	N = 152 UAI: N1 = 105 F/M = 68/37 Mean age: 58.0 ± 11.1 y	BAI: N2 = 47 F/M = 38/9 Mean age: 59.2 ± 10.2 y		
Vinolas 2018 [72] Cross-sectional study	N = 110 N1 = 53 with CS F/M = 42/11 Mean age: 49.9 ± 12.8 y	N2 = 39 with MACS F/M = 34/5 Mean age: 57.8 ± 9.3 y	N3 = 18 with NFAI F/M = 13/5 Mean age: 59.2 ± 9.1 y	
Kim 2018 [73] Cross-sectional study	N = 435 Men (N = 247): N1 = 4 adrenal CS Mean age: 47.5 ± 6.6 y N2 = 30 with SH Mean age: 59.5 ± 8.4 y N3 = 213 with NFAI Mean age: 54.5 ± 9.8 y	Women (N = 188): N1 = 15 adrenal CS Mean age: 41.9 ± 9.7 y N2 = 31 with SH Mean age: 51.2 ± 13.3 y N3 = 142 with NFAI Mean age: 55.4 ± 10.8 y		
Ahn 2019 [74] Cross-sectional study	N = 795 Premenopausal women (N = 77): N1 = 18 with SH Mean age: 38.6 ± 6.1 y N2 = 59 NFAI Mean age: 42.5 ± 6.2 y	Postmenopausal women (N = 237): N1 = 38 with SH Mean age: 59.7 ± 7.4 y N2 = 199 NFAI Mean age: 59.4 ± 7.4 y	Men (N = 481): N1 = 53 with SH Mean age: 56.9 ± 9.7 y N2 = 428 NFAI Mean age: 55.3 ± 9.6 y	
Moraes 2020 [75] Cross-sectional study	N = 75 N1 = 30 ACS F/M = 26/4 Median age: 60 y (42–77 y)	N2 = 45 NFAI F/M = 32/13 Median age: 59 y (32–76 y)		
Athimulam 2020 [76] Cross-sectional study	N = 213 N1 = 22 with CS F/M = 18/4 Mean age: 41.5 y (18–61 y)	N2 = 92 with MACS F/M = 57/35 Mean age: 59.5 y (28–82 y)	N3 = 99 with NFAI F/M = 67/32 Mean age: 59 y (28–93 y)	
Ueland 2020 [77] Retrospective study	N = 165 N1 = 83 with ACS F/M = 58/25 Median age: 65 y (29–86 y)	N2 = 82 with NFAI F/M = 48/34 Median age: 68.5 y (33–82 y)		
Ishida 2021 [78] Retrospective study	N = 67 N1 = 55 with ACS F/M = 33/23 Mean age: 61.5 ± 10.1 y	N2 = 12 with NFAI F/M = 5/7 Mean age: 66.0 ± 8.9 y		
Podbregar 2021 [79] Prospective study	N = 67 with NFAI F/M = 47/20 Mean age: 57.9 y			
Li 2021 [80] Cohort study	N1 = 1004 with AI F/M = 582/422 Mean age: 62.8 y (20.5–96.4 y)	N2 = 1004 controls F/M = 582/422 Mean age: 62.7 y (20.5–95.5 y)		
Yano 2022 [81] Cross-sectional study	N = 194 N1= 97 with ACS F/M = 60/37 Median age: 62.0 y	N2 = 97 with NFAI F/M = 52/45 Median age: 58.0 y		
Izawa 2022 [82] Cross-sectional observational study	N = 237 with adrenal adenoma associated with cortisol excess N1 = 112 with osteoporosis/osteopenia F/M = 99/13 Median age: 57 y (42–64 y)	N2= 125 without osteoporosis/osteopenia F/M = 83/42 Median age: 55 y (46–63 y)		

Abbreviations: ACS = autonomous cortisol secretion; AI = adrenal incidentaloma; ACTH = adrenocorticotropic hormone; BAI = bilateral adrenal incidentaloma; CRH = corticotropin-releasing hormone; DST = dexamethasone suppression test; LDDST = low dose dexamethasone suppression test; F = female; HPA = hypothalamic–pituitary–adrenal; N = number of patients; NFAI = non-functioning adrenal incidentaloma; M = male; MACS = mild autonomous cortisol secretion; SCS = subclinical Cushing syndrome; SH = subclinical hypercortisolism; UFC = urinary free cortisol; UAI = unilateral AI; VFx = vertebral fractures; y = year; (red = SCS or SH subgroups; blue = AI or NFAI subgroup); green colour means CS; blue colour means AI/NFAI; red colour means ACS, MACS, SCS or preclinical SCS.

**Table 2 jcm-12-04244-t002:** Included studies and criteria for ACS (and equivalent) diagnosis [43,44,45,46,47,48,49,50,51,52,53,54,55,56,57,58,59,60,61,62,63,64,65,66,67,68,69,70,71,72,73,74,75,76,77,78,79,80,81,82].

First Author Year of Publication Reference Number Study Design	Criteria for ACS Diagnosis
Reincke 1992 [43] Prospective study	Serum cortisol after 1 mg and 8 mg DST > 90 nmol/L(3 µg/dL)
Ambrosi 1995 [44] Case-control study	Serum cortisol after 1 mg DST and loperamide test (16 mg) or LDDST or 8 mg overnight > 140 nmol/L
Osella 1997 [45] Case-control study	Serum cortisol after 1 mg DST < 2.5 µg/dL (vs. CS)
Sartorio 1998 [46] Case-control study	Serum cortisol after 1 mg DST and loperamide test (16 mg) > 5 µg/dL (140 nmol/L) Low ACTH High UFC Impaired response to CRH test
Rossi 1999 [47] Prospective study	≥2 out of: Serum cortisol after 2 mg DST or LDDST > 3 µg/dL Low ACTH High UFC High F rhythm
Torlontano 1999 [48] Case-control study	UFC > 70 μg/24 h
Osella 2001 [49] Case-control study	≥2 out of: Serum cortisol after 1 mg DST > 5 µg/dL Low ACTH High UFC > 216 µg/24 h High F rhythm
Chiodini 2001 [50] Longitudinal study	≥2 out of: Serum cortisol after 1 mg DST > 82.8 nmol/L Low ACTH < 2.2 pmol/L High UFC > 193.1 nmol/24 h
Tauchmanova 2001 [51] Case-control study	Serum cortisol after LDDST > 3 µg/dL(83 nmol/L)
Chiodini 2002 [52] Cross-sectional study	≥2 out of: Serum cortisol after 1 mg DST > 82.8 nmol/L (3 µg/dL) Low ACTH < 2.2 pmol/L High UFC > 193.1 nmol/24 h
Francucci 2002 [53] Case-control study	N2 included 2 patients with SCS (inclusion criteria: two altered parameters of HPA function without evident clinical signs of hypercortisolism)
Hadjidakis 2003 [54] Cross-sectional study	Serum cortisol after LDDST > 70 nmol/L
Chiodini 2004 [55] Case-control study	≥2 out of: Serum cortisol after 1 mg DST > 3 µg/dL (82.8 nmol/L) Low ACTH < 10 pg/mL (2.2 pmol/L) High UFC > 70.0 µg/24 h (193.1 nmol/24 h)
Tauchmanova 2007 [56] Case-control study	Serum cortisol after 1 mg DST > 30 ng/mL
Tauchmanova 2009 [57] Case-control study	≥2 out of: Serum cortisol after 1mg DST > 30 ng/mL Abnormalities of ACTH Anomalies of UFC Anomalies of F rhythm
Chiodini 2009 [58] Retrospective study	≥2 out of: Serum cortisol after 1 mg DST > 82.8 nmol/L (>3 µg/dL) Low ACTH 1 pg/mL (<2.2 pmol/L) High UFC > 70 µg/24 h (193.1 nmol/L)
Chiodini 2009 [59] Case-control study	≥2 out of: Serum cortisol after 1 mg DST >82.8 nmol/L (>3 µg/dL) Low ACTH < 2.2 pmol/L High UFC > 193.1 nmol/L
Toniato 2009 [60] Prospective study	Serum cortisol after 1 mg DST > 2.5 µg/dL
Morelli 2010 [61] Retrospective study	≥2 out of: Serum cortisol after 1 mg DST > 82.8 nmol/L high UFC Low ACTH < 2.2 pmol/L
Morelli 2011 [62] Cross-sectional study	≥2 out of: Serum cortisol 1mg DST > 3 µg/dL(>83 nmol/L) Low ACTH < 10 pg/mL (<2.2 pmol/L) High UFC > 70 µg/24 h (>193 nmol/L)
Eller-Vainicher 2012 [63] Case-control study	≥2 out of: Serum cortisol after 1 mg DST > 3.0 mg/dL (82.8 nmol/L) Low ACTH < 10 pg/ mL (2.2 pmol/L) High UFC > 70 µg/24 h (193.1 nmol/L)
Iacobone 2012 [64] Prospective study	Serum cortisol after 1 mg DST > 5 µg/dL Low ACTH < 10 pg/mL High UFC > 76 µg/24 h
Morelli 2013 [65] Prospective study	≥2 out of: Serum cortisol after 1 mg DST > 83 nmol/L High UFC > 193 nmol/24 h Low ACTH < 2.2 pmol/L
Palmieri 2013 [66] Cross-sectional, prospective study	≥2 out of: Serum cortisol after 1 mg DST > 83 nmol/L High UFC > 193 nmol/24 h Low ACTH < 2.2 pmol/L
Lasco 2014 [67] Cross-sectional study	Serum cortisol after 1 mg DST > 1.8 µg/dL (50 nmol/L), in case of no suppression preform LDDST > 1.8 µg/dL (50 nmol/L) High UFC > 193 nmol/24 h
Perogamvros 2015 [68] Retrospective study	Serum cortisol after LDDST > 1.8 µg/dL+ at least one of: Low ACTH < 10 pg/mL Midnight serum cortisol > 7 µg/dL High UFC > 120 µg/24 h
Morelli 2016 [69] Retrospective (cross-sectional arm) and prospective (longitudinal arm)	Serum cortisol after 1 mg DST ≥ 2 µg/dL (55 nmol/L)
Salcuni 2016 [70] Cross-sectional and prospective interventional study	Serum cortisol after 1 mg DST > 5.0 µg/dL (138 nmol/L) or ≥2 out of: Serum cortisol after 1 mg DST > 3.0 µg/dL (83 nmol/L) Low ACTH < 10 pg/mL (2.2 pmol/L) High UFC > 70 µg/24 h(193 nmol/24 h)
Ognjanovic 2016 [71] Cross-sectional study	≥2 out of: Serum cortisol after 1 mg DST > 50 nmol/L Serum cortisol after LDDST 50 > nmol/L Midnight serum cortisol > 208 nmol/L Low ACTH < 10 pg/mL (2.2 pmol/L) High UFC > 245 nmol/24 h
Vinolas 2018 [72] Cross-sectional study	Serum cortisol after 1 mg DST > 50 nmol/L
Kim 2018 [73] Cross-sectional study	Serum cortisol after 1 mg DST > 5.0 μg/dL (138 nmol/L) or Serum cortisol after 1 mg DST > 2.2 μg/dL (61nmol/L) plus one of: Low ACTH < 10 pg/mL (2.2 pmol/L) or DHEA-S < 80 μg/dL (2.17 μmol/L) in men/<35 μg/dL (0.95 μmol/L) in women
Ahn 2019 [74] Cross-sectional study	Serum cortisol after 1 mg DST > 138.0 nmol/L or Cortisol after 1 mg DST > 61.0 nmol/L plus: ACTH < 2.2 pmol/L or DHEA-S < 2.17 μmol/L in men or <0.95 μmol/L in women
Moraes 2020 [75] Cross-sectional study	Serum cortisol after 1 mg DST 1.9–5 µg/dL
Athimulam 2020 [76] Cross-sectional study	Serum cortisol after 1 mg DST cortisol > 1.8 µg/dL (>50 nmol/L)
Ueland 2020 [77] Retrospective study	Serum cortisol after 1 mg DST cortisol > 50 nmol/L A proposed cut-off for DHEAS in the diagnostics of ACS is 1.04 µmol/L (40 µg/dL)
Ishida 2021 [78] Retrospective study	Serum cortisol after 1 mg DST > 1.8 µg/dL or 8 mg DST ≥ 1.0 μg/dL
Podbregar 2021 [79] Prospective study	Serum cortisol after 1 mg DST > 50 nmol/L
Li 2021 [80] Cohort study	Serum cortisol after 1 mg DST > 1.8 µg/dL During the follow-up period, 15 (22.4%) of 67 patients with NFAI progressed to MACS (*p* < 0.001)
Yano 2022 [81] Cross-sectional study	Serum cortisol after 1 mg DST ≥ 1.8 μg/dL (50 nmol/L)
Izawa 2022 [82] Cross-sectional observational study	Serum cortisol after 1 mg DST cortisol levels > 1.8 µg/dL

Abbreviations: ACS = autonomous cortisol secretion; AI = adrenal incidentaloma; ACTH = adrenocorticotropic hormone; BAI = bilateral adrenal incidentaloma; CRH = corticotropin-releasing hormone; DST = dexamethasone suppression test; LDDST = low dose dexamethasone suppression test; F = female; HPA = hypothalamic-pituitary-adrenal; N = number of patients; NFAI = non-functioning adrenal incidentaloma; M = male; MACS = mild autonomous cortisol secretion; SCS = subclinical Cushing syndrome; SH = subclinical hypercortisolism; UFC = urinary free cortisol; UAI = unilateral AI; VFx = vertebral fractures; y = year.

**Table 3 jcm-12-04244-t003:** DXA results in terms of BMD assessment and/or osteoporosis/osteopenia prevalence in included studies (to identify each sample size/subgroup, please check Table 1); the display starts from the oldest data to the most recent [43,47,48,49,50,51,52,53,54,55,56,57,58,59,60,61,62,63,64,65,66,67,68,69,70,71,72,73,74,75,76,77,78,79,80,81,82].

Reference Number	DXA Results Osteoporosis/VFs Prevalence
[43]	Osteoporosis prevalence: N1 = 37.5%; N2 = 0%; N3 = 0%
[47]	BMD Z-score similar between N1 and N3 (−0.41 SD, 95% CI: 21.127–0.3115)
[48]	N1:QCT (L1–L4) = −1.02 ±1.29; DXA (L2–L4) = −1.37 ± 1.26; FN = −1.34 ± 1.08 N2: QCT (L1–L4) = 0.15 ± 1.04; DXA (L2–L4) = 0.41 ± 1.24; FN = 0.20 ± 1.12 N3: QCT (L1–L4) = −0.20 ± 0.86; DXA (L2–L4) = −0.03 ± 1.07; FN = 0.07 ± 1.05 BMD at each site was lower in group N1 than in group N2 and N3 (*p* < 0.05)
[49]	N1: LS (L1–L4) BMD = 0.926 g/cm^2^ (range: 0.604–1.144); LS BMD values were similar between N1 and N3 (*p* = NS) T-score = −1.10 SD (range: −4.17–0.88); Z-score = −0.33 SD (range: −3.86–1.79) N2: LS (L1–L4) BMD = 0.929 g/cm^2^ (range: 0.707–1.144); Z-score = −0.55 SD (range: −1.82–1.79) N3: LS (L1–L4) BMD = 0.936 g/cm^2^ (range: 0.645–1.268) T-score = −1.10 SD (range: −3.65–2.01); Z-score = −0.34 SD (range: −2.96–2.67)
[50]	(Z-score) N1:DXA (L2–L4) = −0.67 ± 0.32 SD; QCT (L1–L4) = −0.71 ± 0.63; FN = −0.37 ± 0.6 SD N2: DXA (L2–L4) = 0.64 ± 1.3 SD; QCT (L1–L4) = 0.26 ± 0.98; FN = 0.35 ± 1.0 SD BMD measured by both QCT (L1–L4) and DXA (L2–L4) was significantly lower in N1 vs. N2 (*p* < 0.05)
[51]	N1: Proximal finger Ad-SoS = 1941 ± 126 m/s LS BMD = 0.87 ± 0.15 g/cm^2^; LS (Z-score) = −1.4 ± 1.4 SD FN BMD = 0.75 ± 0.12 g/cm^2^ 2 patients with VFx N2: Proximal finger Ad-SoS = 1968 ± 139 m/s LS BMD = 0.93 ± 0.16 g/cm^2^; LS (Z-score)= −0.86 ± 1.3 SD FN BMD = 0.83 ± 0.09 g/cm^2^ 3 patients with VFx N3: Proximal finger Ad-SoS = 2103 ± 43 m/s LS BMD = 1.05 ± 0.1 g/cm^2^; LS (Z-score)= −0.14 ± 0.05 SD FN BMD = 0.93 ± 0.105 g/cm^2^ Significant bone loss was detected by finger Ad-SoS in N1 and N2 vs. N3 (*p* < 0.001)
[52]	N1: DXA L2–L4 (Z-score) = −0.42 ± 1.62 SD (LS BMD Z-score lower in N1 vs. N2 vs. N3, *p* < 0.05) FN BMD (Z-score) = 0.02 ± 1.19 SD N2: DXA L2–L4 (Z-score) = 0.60 ± 1.13 SD FN BMD (Z-score) = 0.70 ± 0.95 SD N3: DXA L2–L4 (Z-score) = 0.47 ± 1.06 SD FN BMD (Z-score) = 0.49 ± 1.11 SD Prevalence of osteoporosis/osteopenia: higher in N1 vs. N2 (84.6% vs. 36.0%, *p* = 0.01)
[53]	LS and FN BMD Z-score: lower in N1 vs. N2 and N3 (*p* < 0.05)
[54]	N1: L2–L4 BMD = 0.89 ± 0.14 g/cm^2^: No difference in L2–L4 (BMD and Z-score) between N1 and N2 (*p* = 0.78, *p* = 0.36) L2–L4 (Z-score)= 0.51 ± 1.2 SD FN BMD = 0.72 ± 0.09 g/cm^2^; FN (Z-score) = −0.2 ± 0.08 SD (FN BMD was significantly lower in N1 vs. N2; *p* < 0.05) N2: L2–L4 BMD = 0.9 ± 0.16 g/cm^2^ L2–L4 (Z-score) = 0.11 ± 1.5 SD FN BMD = 0.78 ± 0.1 g/cm^2^; FN (Z-score) = 0.43 ± 0.94 SD
[55]	QCT (L1–L4) (Z-score) Premenopause: N1:−0.11 ± 0.48 (−1.91–1.77); N2: 0.57 ± 0.34 (−1.64–2.57); N3: 0.13 ± 0.26 (−1.95–2.33) Postmenopause: N1: −0.78 ± 0.29 (−2.49–1.24); N2: −0.02 ± 0.19 (−2.49–1.24); N3: 0.06 ± 0.14 (−1.96–2.20) VFs prevalence: Premenopause: N1 = 42.9%; N2 = 7.1%; N3 = 0% Postmenopause: N1 = 78.6%; N2 = 42.9%; N3 = 37.7% The prevalence of fractures was higher in SH from both groups: premenopause (N1 vs. N3, *p* = 0.001; N2 *p* = 0.049) and postmenopause (N1 vs. N2, *p* = 0.006; N2 *p* = 0.024); in the postmenopausal group, Z-score was lower in N1 than N3 (*p* = 0.011) and N2 (*p* = 0.034)
[56]	N: Lumbar Z-score = −1.8 SD (−5.1 to −0.37); Femoral Z-score= −0.9 SD (−2.35 to −0.29) N2: Lumbar Z-score = −1.1 SD (−4.2 to −0.35); Femoral Z-score = −0.6 SD (−1.8 to −0.25) N3: Lumbar Z-score = −0.02 SD(−0.7 to −1.1); Femoral Z-score SD = 0.05 (−0.3 to −0.4) VFx prevalence: N1 = 69% (multiple in 36% of them); N2 = 57% (multiple in 31%); N3 = 0 VFx (between N1 and N2: *p* = 0.56) LS BMD values and cortisol-to-DHEAS ratio were the best predictors of VFx (*p* < 0.01)
[57]	N1: LS BMD = 0.97 ± 0.12 g/cm^2^; FN BMD = 0.816 ± 0.14 g/cm^2^ LS BMD:N1 < N2 (*p* = 0.04) N2: LS BMD = 0.98 ± 0.13 g/cm^2^; FN BMD = 0.817 ± 0.11 g/cm^2^ Incidental fractures appeared only in the untreated group
[58]	N1: LS BMD (Z-score) = −0.73 ± 1.43 SD (−4.5–3.08); FN BMD (Z-score) = −0.37 ± 1.06 SD (−2.5–2.19); SDI = 1.35 ± 1.27(0–7) N3: LS BMD (Z-score) = 0.12 ± 1.21 SD (−2.34–3.04); FN BMD (Z-score) = 0.17 ± 1.02 SD(−2.28–3.65); SDI = 0.31 ± 0.68 (0–4) VFx prevalence:N1 = 70.6%;N2 = 21.8%; N3 = 22.2% (prevalence of VFx and SDI was higher in N1 than in N2 and N3, *p* < 0.0001) VFx and SDI were associated with SH (OR = 7.27, 95% CI 3.94–13.41, *p* = 0.0001)
[59]	N1: LS BMD (Z-score) = −1.04 ± 1.84 SD (−4.50–4.10); FN BMD (Z-score) = −0.63 ± 1.01 SD (−2.5–1.5) N2: LS BMD (Z-score) = 0.19 ± 1.34 SD (−2.80–3.60); FN BMD (Z-score) = 0.01 ± 1.01 SD (−2.8–2.1) N3: LS BMD (Z-score) = 0.20 ± 1.28 SD (−2.30–2.90); FN BMD (Z-score) = 0.26 ± 1.06 SD (−2.30–2.90) Prevalence of osteoporosis: N1 = 40.9%; N2 = 18.2%; N3 = 16.7% (LS BMD: lower in N1 vs. N2 vs. N3 (*p* = 0.001) and FN (*p* = 0.002) Prevalence of VFx: N1 = 72.7%; N2 = 21.2%; N3 = 20% (N1> vs. N2 vs. N3, *p* = 0.0001) SH was associated with LS BMD (β = −0.378, *p* = 0.0001) and VFx (OR = 7.81, 95% CI 1.96–31.17, *p* = 0.004)
[60]	N1: DXA (T-score) = −1.81 ± 2.07 SD vs. N2: DXA (T-score) = −1.86 ± 1.93 SD\ Prevalence of osteoporosis: N1 = 21.7% vs. N2 = 27.3%
[61]	VFx: 35.1%
[62]	Baseline: N1: LS BMD (Z-score) = 0.01 ± 1.17 SD (−1.8–2.5); FN BMD (Z-score) = −0.04 ± 0.99 SD (−2.4–2.7); SDI = 1.11 ± 1.50 (0–6) N2: LS BMD (Z-score) = 0.03 ± 1.38 SD (−2.8–4.1); FN BMD (Z-score) = 0.07 ± 0.78 SD (−1.6–2.1); SDI = 0.58 ± 1.10 (0–4) After 24 months of follow-up: N1: LS BMD (Z-score) = 0.27 ± 1.37 SD (−2.0–3.6); FN BMD (Z-score) = 0.00 ± 1.07 SD (−2.4–2.6); SDI = 2.11 ± 1.85(0–8) N2: LS BMD (Z-score) = 0.16 ± 1.45 SD (−2.6–4.6); FN BMD (Z-score) = 0.11 ± 0.83 SD (−1.7–2.9); SDI = 0.79 ± 1.40 (0–6) Baseline prevalence of VFx: N1 = 55.6% vs. N2 = 28.9% After follow-up: prevalence of VFx: N1 = 81.5% (New VFx = 48.1%) vs. N2 = 35.5% (New VFx = 13.2%) The risk of developing new VFx was independently associated with the presence of SH (OR = 12.3; 95% CI 4.1–36.5, *p* = 0.001)
[63]	N1: LS BMD (Z-score) = −0.31 ± 1.17 SD (−2.6–1.81); FN BMD (Z-score) = 0.37 ± 0.78 SD (−1.62–1.74) N2: LS BMD (Z-score) = 0.31 ± 1.42 SD (−3.71–3.61); FN BMD (Z-score) = −0.04 ± 0.99 SD (−2.61–2.71) N3: LS BMD (Z-score) = 0.42 ± 1.52(−2.0–3.72); FN BMD (Z-score) = −0.03 ± 0.72 SD (−1.4–1.6) VFx prevalence: N1 = 82.4% vs.N2 = 45.6% vs. N3 = NA LS BMD: N1 < N2 (*p* = 0.03) and vs. N3 (*p* = 0.02)
[64]	N1: Before adrenalectomy LS (T-score) = −1.23 ± 0.74 SD; after adrenalectomy LS(T-score) = −1.29 ± 0.77 SD N2: At baseline LS (T-score) = −1.19 ± 0.74 SD; after follow-up: LS(T-score) = −1.27 ± 0.82 SD Prevalence of osteoporosis/osteopenia: N1: Before adrenalectomy = 30%; after adrenalectomy: 1 osteopenic patient became osteoporotic N2: At baseline: = 26.7%; after follow-up: 1 osteopenic patient became osteoporotic There were no changes in T-score in either group (*p* = NS)
[65]	BAI: N1: LS BMD (Z-score) = −0.66 ± 0.45 SD (−2.4–1.3); FN BMD (Z-score) = −0.6 ± 0.3 SD (−2–1) N2: LS BMD (Z-score) = −0.09 ± 0.26 SD (−2.8–3.5); FN BMD (Z-score) = −0.28 ± 0.2 SD (−2–1.2) UAI: N1: LS BMD (Z-score) = −0.25 ± 0.22 SD (−3.6–2.9); FN BMD (Z-score) = −0.17 ± 0.2 SD (−2.8–2.5) N2: LS BMD (Z-score) = 0.23 ± 0.12 SD (−2.8–5.7; FN BMD (Z-score) = 0.14 ± 0.1 SD (−2.6–4.3) Prevalence of VFX: BAI (N1 = 70% vs.N2 = 46.4%); UAI (N1 = 46.3% vs.N2 = 24.6%) Presence of VFx was associated with BAI after adjusting for SH (OR = 1.77, 95% CI 0.85–3.7, *p* = 0.12) and LS BMD (OR = 1.31, 95% CI 1.03–1.67, *p* = 0.03)
[66]	N1: LS BMD (Z-score) = −0.66 ± 1.4 SD (−3–3); FN BMD (Z-score) = −1.03 ± 1.1 SD (−2.1–2.2) N2: LS BMD (Z-score) = 0.81 ± 1.5 SD (−3–4); FN BMD (Z-score) = 0.6 ± 0.9 SD (−1.6–2.4) Prevalence of osteoporosis: N1 = 87.5% vs. N2 = 27.8% (*p* < 0.0001) Prevalence of VFx: N1 = 68.8% vs. N2 = 18.5% (*p* < 0.0001)
[67]	N1: LS BMD = 0.92 ± 0.04 g/cm^2^; LS BMD (T-score) = −0.76 ± 0.15 SD; FN BMD = 0.70 ± 0.04 g/cm^2^; FN BMD (T-score) = −1.33 ± 0.30 SD N2: LS BMD = 0.76 ± 0.10 g/cm^2^: LS BMD (T-score) = −2.90 ± 0.95 SD; FN BMD = 0.67 ± 0.08 g/cm^2^: FN BMD (T-score) = −1.60 ± 0.60 SD Prevalence of VFx: N1 = 100% vs. N2 = 10.6% VFx (SH was associated with a high prevalence of VFx independent of BMD)
[68]	N1 = 3 patients with osteoporosis vs. N2 = 5 patients with osteoporosis
[69]	N1: LS BMD (Z-score) = −0.02 ± 1.29 SD (−2.8–4.1) FN BMD (Z-score) = 0.06 ± 0.75 SD (−1.6–2.1) N2: LS BMD (Z-score) = 0.37 ± 1.29 SD (2–2.7) FN BMD (Z-score) = −0.13 ± 1.16 SD (−2.4–2.7) VFx rate: 33% (N1) vs. 53.3% (N2) Risk of new VFx at diagnosis and during follow-up (OR = 10.27, 95% CI 3.39–31.12, *p* = 0.0001)
[70]	N1: Baseline: LS BMD (Z-score) = −0.86 ± 1.27 SD (−2.9–1.9); FN BMD (Z-score) = −0.54 ± 0.9 SD (−1.9–1.7) Follow-up: LS BMD (Z-score) = −0.49 ± 1.17 SD (−2.7–2.1); LS ΔZ-score/year = 0.10 ± 0.2 SD (0.0–1.0) FN BMD (Z-score) = −0.41 ± 0.9 SD (−1.7–1.8); FN ΔZ-score/year = −0.09 ± 2.0 SD (−4.26–4.76) N2: Baseline: LS BMD (Z-score) = 0.23 ± 1.4 SD (−1.8–2.7); FN BMD (Z-score) = 0.14 ± 1.2 SD (−2.4–2.7) Follow-up: LS BMD (Z-score) = 0.25 ± 1.5 SD (−2.1–2.8); LS ΔZ-score/year = −0.01 ± 0.3 SD (0.03–0.17) FN BMD (Z-score) = 0.12 ± 1.2 SD (−2.4–2.6); FN ΔZ-score/year = −0.58 ± 2.3 SD (−0.83–0.64) Prevalence of VFX: N1: baseline = 46.9%; follow-up = 46.9% (new VFx = 9.4%) N2: baseline = 65.2%; follow-up = 91.3% (new VFx = 52.2%) LS Z-score (ΔZ-score/year) tended to increase in N1 (0.10 ± 0.20 SD) compared with N2 (−0.01 ± 0.27 SD, *p* = 0.08) Surgery in AI patients with SH was associated with a 30% VFx risk reduction (OR = 0.7, 95% CI 0.01–0.05, *p* = 0.008)
[71]	N1: LS BMD = 0.96 ± 0.14 g/cm^2^ LS BMD was lower in BAI than in UAI patients (0.96 ± 0.14 vs. 0.87 ± 0.15 g/cm^2^, *p* = 0.002) FN BMD = 0.74 ± 0.11 g/cm^2^ hip BMD = 0.89 ± 0.13 g/cm^2^ N2: LS BMD = 0.87 ± 0.15 g/cm^2^ FN BMD = 0.70 ± 0.12 g/cm^2^ hip BMD = 0.85 ± 0.13 g/cm^2^ Prevalence of osteoporosis: N1 = 15.9% vs. N2 = 37.1% (*p* = 0.011)
[72]	N2: LS BMD (T-score) = 1.06 ± 0.20 SD N3: LS BMD (T-score) = 1.11 ± 0.18 SD
[73]	VFx prevalence: men—N1 = 25.0%; N2 = 0%; N3 = 1.2%; women—N1 = 0.5%; N2 = 0%; N3 = 3.2%
[74]	Premenopausal women: N1: LS BMD = 0.993 ± 0.131 g/cm^2^; FN BMD = 0.812 ± 0.107 g/cm^2^; FT BMD = 0.865 ± 0.096 g/cm^2^ N2: LS BMD = 1.108 ± 0.160 g/cm^2^; FN BMD = 0.900 ± 0.153 g/cm^2^; FT BMD = 0.972 ± 0.137 0.865 g/cm^2^ Postmenopausal women: N1: LS BMD = 0.920 ± 0.146 g/cm^2^; FN BMD = 0.775 ± 0.127 g/cm^2^; FT BMD = 0.863 ± 0.104 g/cm^2^ N2: LS BMD = 0.993 ± 0.172 g/cm^2^; FN BMD = 0.797 ± 0.130 g/cm^2^; FT BMD = 0.887 ± 0.148 g/cm^2^ Men: N1: LS BMD = 1.088 ± 0.188 g/cm^2^; FN BMD = 0.87± 0.135 g/cm^2^; FT BMD = 0.964 ± 0.127 g/cm^2^ N2: LS BMD = 1.107 ± 0.186 g/cm^2^; FN BMD = 0.898 ± 0.144 g/cm^2^; FT BMD = 1.01± 0.150 g/cm^2^ Premenopausal women with SH < NFAI: lower LS BMD (*p* = 0.008), FN (*p* = 0.012), FT (*p* = 0.009) Postmenopausal women with SH > NFAI: lower LS BMD (*p* = 0.016) Prevalence of VFx: (premenopause) N1 = 0% vs.N2 = 1.7%; (postmenopause) N1 = 0% vs.N2 = 4%; (men) N1 = 0% vs. N2 = 0.7% Similar VFx rates in premenopausal women with SH and NFAI (*p* = 0.578), postmenopausal women with SH and NFAI (*p* = 0.209), or men with SH and NFAI (*p* > 0.999)
[75]	N1: LS BMD = 1.007 g/cm^2^ (0.861–1.314); FN aBMD = 0.917 g/cm^2^ (0.766–1.170); 33% Radius aBMD = 0.626 g/cm^2^ (0.496–0.935) N2:LS BMD = 1.125 g/cm^2^ (0.793–1.681); FN aBMD = 1.002 g/cm^2^ (0.710–1.384); 33% Radius aBMD = 0.706 g/cm^2^ (0.528–0.890) Prevalence of osteoporosis: N1 = 75% vs. N2 = 64.9% Prevalence of VFx: N1 = 73.7% vs. N2 = 55.6% (N1 vs. N2, *p* = 0.24)
[76]	Prevalence of osteopenia and osteoporosis: N1: 62%; 38%; N2: 56%; 21%; N3: 52%;19%
[77]	Prevalence of osteoporosis: N1 = 18.1% vs.N2 = 8.5%
[78]	N1: LS BMD (Z-score) = 0.33 ± 1.2 SD; FN BMD (Z-score) = −0.38 ± 1.0 SD N2: LS BMD (Z-score) = 0.90 ± 1.17 SD; FN BMD (Z-score) = 0.23 ± 1.20 SD Prevalence of VFx: N1 = 63.6% vs. N2 = 70.0% (UFC was higher in mild N1 with VFx than in those without VFx, *p* = 0.037)
[79]	Baseline: 17.9% had osteoporosis → Follow-up (10.5 y): 26.9% had osteoporosis (*p* = 0.031)
[80]	N1: LS BMD = 1.1 g/cm^2^ (0.5–1.5); FN BMD = 0.8 g/cm^2^ (0.6–1.3); hip BMD = 0.9 g/cm^2^ (0.59–1.4) N2: LS BMD = 1.0 g/cm^2^ (0.7–1.6); FN BMD = 0.8 g/cm^2^ (0.5–1.2); hip BMD = 0.9 g/cm^2^(0.5–1.3) Prevalence of any osteoporotic fracture: 16.6% vs.13.3% (*p* = 0.04) Prevalence of VFx: 6.4% vs. 3.6% (*p* = 0.004) Cumulative incidence of any new VFx: 10 y = 3.5% vs.10 y = 3.6% (*p* = 0.33) Risk of developing new fractures during follow-up in N1 vs. N2 OR = 1.27 (95% CI: 1.07–1.52)
[81]	N1: LS BMD = 0.87 g/cm^2^ (0.74–1.00); LS BMD (Z-score) = −0.2 SD (−0.9–0.7) FN BMD = 0.62 g/cm^2^ (0.53–0.72); FN BMD (Z-score) = −0.6 SD (−1.2–0.3) N2: LS BMD = 0.88 g/cm^2^ (0.80–1.06); LS BMD (Z-score) = 0.1 SD (−0.6–0.9) FN BMD = 0.65 g/cm^2^ (0.60–0.76); FN BMD (Z-score) = −0.3 SD (−0.8–0.4) Prevalence of VFx: N1 = 49% vs. N2 = 8% (N1 > N2 had higher rates of coexistence of VFx and arterial stiffness: 23% vs. 2%, *p* < 0.001) and of VFx and abdominal aortic calcification: 22% vs. 1%, *p* < 0.001)
[82]	N1: LS BMD% (IQR) = 74 (68–81); FN BMD% (IQR) = 75 (68–79); distal radius BMD %(IQR) = 93 (77–104) N2: LS BMD% (IQR) = 96 (91–105); FN BMD% (IQR) = 94 (86–100); distal radius BMD%(IQR) = 97 (89–106) Prevalence of fragility fractures: N1 = 31.3% vs.N2 = 0% MACS and 1-mg DST cortisol were positively associated with osteoporosis/osteopenia (OR = 1.156, 95% CI: 1.046–1.278, *p* = 0.005)

Abbreviations: ACS = autonomous cortisol secretion; aBMD = areal bone mineral density; Ad-SoS = amplitude dependent speed of sound; AI = adrenal incidentalomas, ALP = alkaline phosphatase; BAI = bilateral adrenal incidentaloma; baPWV = brachial-ankle pulse wave velocity; BMD = bone mineral density; F = female; DHEA-S = dehydroepiandrosterone sulphate; DST = dexamethasone suppression test, DXA = dual energy X-ray absorptiometry; FN = femoral neck; FT = total femur; HR-pQCT = high-resolution peripheral quantitative computed tomography; LS = lumbar spine; M = male; MACS = mild autonomous cortisol secretion; N = number of patients; NFAI = non-functioning adrenal incidentaloma; NS = not significantly; OP = osteopenia; OR = odds ratio; OS = osteoporosis; QCT = quantitative computed tomography; SCS = subclinical Cushing syndrome; SD = standard deviation; SDI = spinal deformity index; SH = subclinical hypercortisolism; TBS = trabecular bone score; UAI = unilateral adrenal incidentaloma, UFC = urinary free cortisol; vBMD = volumetric bone mineral density; VFx = vertebral fractures; vs. = versus; y = years.

**Table 4 jcm-12-04244-t004:** TBS analysis in patients diagnosed with ACS/AIs [63,72,73,81].

Reference Number	TBS Results
[63]	N1: TBS (Z-score) = −3.184 ± 1.211; N2: TBS (Z-score) = −1.704 ± 1.541; N3: TBS (Z-score) = −1.189 ± 0.991 The presence of fracture was associated with low TBS (OR = 4.8; 95% CI: 1.85–12.42, *p* = 0.001) and with the cluster of low TBS plus low LS-BMD (OR = 4.37; 95% CI, 1.71–11.4, *p* = 0.002)
[72]	N2: TBS = 1.30 ± 0.09; N3: TBS = 1.37 ± 0.12 TBS was significantly decreased in N3 vs. N2 (*p* < 0.04), but not BMD (*p* = 0.34) After remission of CS, TBS has improved more markedly and rapidly than BMD (10% vs. 3%, *p* < 0.02)
[73]	1 mg DST—cortisol was inversely correlated with TBS in men (β = −0.133, *p* = 0.045) and women (β = −0.140, *p* = 0.048). Compared with women with NFAI, women with SH had 2.2% lower TBS (*p* = 0.040).
[81]	N1: TBS = 1.34(1.28–1.39); N2: TBS = 1.37(1.31–1.42) baPWV was negatively correlated with TBS (r = −0.33, *p* = 0.002)

Abbreviations: BMD = bone mineral density; baPWV = brachial-ankle pulse wave velocity; CI = confidence interval; DST = dexamethasone suppression test; LS = lumbar spine; N = number of patients; NFAI = non-functioning adrenal incidentaloma; OR = odds ratio; TBS = trabecular bone score.

**Table 5 jcm-12-04244-t005:** BTMs and calcium metabolism evaluations in patients with ACS and AIs; for a description of the studied population, please check Table 1 [44,45,46,48,49,50,51,52,53,54,55,56,57,63,65,70,75,76,78,81,82].

Reference Number	BTMs and Calcium and Mineral Metabolism Assays
[44]	Osteocalcin, ICTP, PIIINP: significantly lower in N1 and N2 vs. N3 (*p* < 0.01)
[45]	N1: Osteocalcin = 3.0 ng/mL (1.1–8.6) PICP = 111.5 µg/L (68–307) ICTP = 4.0 µg/L (1.8–7.9) PIIINP = 2.2 µg/L (1.3–3.7) bone ALP = 4.4 µg/L (1–22.5) N2: Osteocalcin = 6.6 ng/mL (1.8–12.9): Lower osteocalcin in N2 vs. N3 (*p* < 0.05) PICP = 119 µg/L (78–223) ICTP = 4.2 µg/L (1.1–6.3): Higher ICTP in N2 vs. N3 (*p* < 0.01) PIIINP = 3.1 µg/L (2.3–6.2) bone ALP = 12.1 µg/L (7–24) N3: Osteocalcin = 7.8 ng/mL (3.8–17) PICP = 108 µg/L (63–163) ICTP = 3.1 µg/L (1–6.1) PIIINP = 3.5 µg/L (2.1–7.8) bone ALP = 11.5 µg/L (3.1–18.5)
[46]	N1: Osteocalcin = 0.9 ± 0.2 ng/mL ICTP = 2.7 ± 0.2 ng/mL PIIINP = 1.9 ± 0.2 ng/mL N2: Osteocalcin = 2.5 ± 0.8 ng/mL: Osteocalcin, ICTP, PIIINP N2 < N4 (*p* < 0.0001, *p* < 0.0001, *p* < 0.02) ICTP = 2.2 ± 0.1 ng/mL PIIINP = 2.2 ± 0.2 ng/mL N3: Osteocalcin = 4.2 ± 0.5 ng/mL: Osteocalcin, ICTP N3 < N4 (*p* < 0.05 and *p* < 0.001) ICTP = 2.9 ± 0.2 ng/mL PIIINP = 3.6 ± 0.2 ng/mL N4: Osteocalcin = 5.5 ± 0.2 ng/mL ICTP = 3.9 ± 0.2 ng/mL PIIINP = 3.2 ± 0.2 ng/mL
[47]	N1:Osteocalcin = 3.8 ± 2.3 ng/mL: Osteocalcin was lower in N1 vs. N2 vs. N3, (*p* < 0.05) ICTP = 4.08 ± 1.29 µg/L D-Pyr/Cr = 28.6 ± 12.8 pmol/pmoL PTH = 57.1 ± 13.6 pg/mL ALP = 207 ± 104 U/L N2: Osteocalcin = 7.5 ± 3.1 ng/mL ICTP = 3.90 ± 2.39 µg/L D-Pyr/Cr = 24.6 ± 7.9 pmol/pmoL PTH = 46.0 ± 14.8 pg/mL ALP = 190 ± 70 U/L N3: Osteocalcin = 8.8 ± 3.2 ng/mL ICTP = 4.01 ± 1.57 µg/L D-Pyr/Cr = 24.6 ± 6.8 pmol/pmoL PTH = 37.2 ± 10.9 pg/mL ALP = 167 ± 43 U/L PTH was higher in N1 vs. N2 (*p* < 0.05); PTH positively was correlated with D-Pyr/Cr (r = 0.72, *p* < 0.05) and inversely correlated with BMD measured at FN (r = −0.92, *p* < 0.002)
[49]	PTH: Inversely correlated with either LS or FN BMD values (r = −0.5; r^2^ = 0.25; *p* = 0.015, and r = 0.42; r^2^ = 0.18; *p* = 0.03).
[50]	N1: Osteocalcin = 1.22 ± 0.41 pmol/L D-Pyr/Cr = 18.6 ± 6.31 pmol/pmoL PTH = 56.2 ± 10.2 ng/L (10–70) N2: Osteocalcin = 1.41 ± 0.45 pmol/L D-Pyr/Cr = 22.1 ± 8.22 pmol/pmoL (BGP, D-Pyr and D-Pyr/Cr were similar in N1 and N2, *p* = NS) PTH = 45.6 ± 12.5 ng/L (10–70) (PTH tended to be higher in N1 vs. N2 but not statistically significantly, *p* = 0.06)
[51]	N1: Ca = 9.26 ± 0.51 mg/dL Phosphorus = 3.86 ± 0.73 mg/dL ALP = 176 ± 83.7 U/L N2:Ca = 9.23 ± 0.42 mg/dL Phosphorus = 3.46 ± 0.4 mg/dL ALP = 164 ± 39 U/L N3: Ca = 9.10 ± 0.6 mg/dL Phosphorus = 3.7 ± 0.7 mg/dL ALP = 169 ± 48 U/L
[52]	N1: Osteocalcin = 3.2 ± 1.0 pmol/L D-Pyr/Cr = 23.2 ± 21.5 pmol/pmoL PTH = 38.5 ± 10.6 ng/L N2: Osteocalcin = 2.8 ± 0.7 pmol/L D-Pyr/Cr = 13.5 ± 5.9 pmol/pmoL PTH = 40.3 ± 12.3 ng/L N3: Osteocalcin = 3.9 ± 1.2 pmol/L (Osteocalcin and D-Pyr/Cr were similar between groups) D-Pyr/Cr = 12.7 ± 4.7 pmol/pmoL PTH = 40.9 ±13.8 ng/L (PTH levels were similar between groups)
[53]	N1:Phosphorus = 1.063 ± 0.146mmol/L vs. N2: Phosphorus = 1.115 ± 0.126 mmol/L vs. N3: Phosphorus = 1.235 ± 0.214 mmol/L Significant reduction in osteocalcin in N1 vs. N3 (*p* < 0.05) Significant reductions in osteocalcin and serum phosphorus in N1 and N2 vs. N3 (*p* < 0.05)
[54]	N1: Osteocalcin = 18.6 ± 8.6 ng/mL (Osteocalcin: N1 < N2, *p* < 0.01). PTH = 43 ± 15.6 pg/mL N2: Osteocalcin = 26.2 ± 8.1 ng/mL PTH = 41.2 ± 14.8 pg/mL (Similar PTH N1 = N2, *p* = 0.72), PTH correlated with FN BMD, r = −0.46, *p* < 0.05).
[56]	N1: Osteocalcin = 1.9 ± 0.5 ng/mL CTX = 0.69 ± 0.24 ng/mL Ca = 2.36 ± 0.13 mmol/L PTH = 43 ± 12 ng/L ALP = 164 ± 52 U/L N2: Osteocalcin = 3.4 ± 0.9 ng/mL CTX = 0.56 ± 0.22 ng/mL Ca = 2.34 ± 0.14 mmol/L PTH = 38 ± 15 ng/L ALP = 175 ± 61 U/L N3: Osteocalcin = 8.9 ± 2.4 ng/mL CTX = 0.34 ± 0.18 ng/mL Ca = 2.33 ± 0.11 mmol/L PTH = 40 ± 16 ng/L ALP = 180 ± 58 U/L
[57]	N1: Osteocalcin = 15 ± 3 ng/mL CTX = 1520 ± 923 pmol/L bone ALP = 14.2 ± 4 μg/L Ca = 2.35 ± 0.1 mmol/L iPTH = 43 ± 12 ng/L N2:Osteocalcin = 14.6 ± 3.2 ng/mL (osteocalcin, CTX, and bone ALP decreased in N1 vs. N2, *p* < 0.05) CTX = 1590 ± 1065 pmol/L Bone ALP = 15 ± 3.5 μg/L Ca = 2.34 ± 0.1 mmol/L iPTH = 38 ± 15 ng/L
[63]	N: ALP = 69.1 ± 20.0 U/L; 25OHD = 27.1 ± 14.4 ng/mL vs. N2: ALP = 70.7 ± 19.5U/L; 25OHD = 25.9 ± 13.6 ng/mL
[65]	BAI: N1: 25OHD = 46.3 ± 16.3 nmol/L Ca = 2.3 ± 0.1 mmol/L PTH = 58.3 ± 11 pg/mL ALP = 74.8 ± 22.9 U/L N2: 25OHD = 55 ± 29 nmol/L Ca = 2.3 ± 0.2 mmol/L PTH = 56.8 ± 21.2 pg/mL ALP = 74.1 ± 24.3 U/L UAI: N1: 25OHD = 41.8 ± 26.8 nmol/L Ca = 2.3 ± 0.1 mmol/L PTH = 53.9 ± 19.9p g/mL ALP = 72.8 ± 19.8 U/L N2: 25OHD = 48.3 ± 28.5 nmol/L Ca = 2.3 ± 0.1 mmol/L PTH = 53.4 ± 21.4 pg/mL ALP = 73.1 ± 19.4 U/L
[70]	N1: Baseline: 25OHD = 21.9 ±8.7 ng/mL vs. Follow-up: 25OHD = 39.4 ± 8.1 ng/mL N2: Baseline: 25OHD = 22.4 ±7.3 ng/mL vs. Follow-up: 25OHD = 38.7 ± 7.5 ng/mL
[75]	N1: 25OHD = 24 ng/mL vs. N2: 25OHD = 25 ng/mL
[76]	N1: Osteocalcin = 14.8 ng/mL P1NP = 34.8 µg/L CTX = 0.3 ng/mL Sclerostin = 419 pg/mL N2: Osteocalcin = 20.1 ng/mL P1NP = 48.7 µg/L CTX = 0.4 ng/mL Sclerostin = 538 pg/mL N3: Osteocalcin = 33 ng/mL (Osteocalcin, P1NP N1 < N2 vs. N3, *p* < 0.001), and P1NP, *p* = 0.003). P1NP = 48.5 µg/L CTX = 0.4 ng/mL (similar CTX N1, N2, N3) Sclerostin = 624 pg/mL (sclerostin N1 vs. N2 vs. N3, *p* < 0.0001)
[78]	N1: Osteocalcin = 14.9 ± 7.4 ng/mL U-NTX = 50.6 ± 25.6 nmol BCE/mmol Cr bone ALP = 15.7 ± 6.0 µg/L 25OHD = 16.9 ± 6.8 ng/mL PTH = 45.6 ± 15.3 pg/mL Ca = 9.2 ± 0.40 pg/mL N2: Osteocalcin = 10.3 ± 5.6 ng/mL U-NTX = 26.9 ± 16.6 nmol BCE/mmol Cr (U-NTX: N1 > N2, *p* = 0.017) and bone ALP, *p* = 0.016) bone ALP = 11.1 ± 4.3 µg/L 25-OHD = 18.0 ± 8.2 ng/mL PTH = 48.8 ± 14.4 pg/mL Ca = 9.2 ± 0.41 pg/mL
[81]	N1: Ca = 9.3 mg/dL (9.1–9.5) 25OHD = 13.7 ng/mL (9.7–17.5) PTH = 53.8 pg/mL (38.3–69.2) bone ALP = 12.3 μg/L (9.8–15.4) N2: Ca = 9.3 mg/dL (9.1–9.5) 25OHD = 14.3 ng/mL (10.5–18.5) PTH = 43.8 pg/mL (34.5-54.7) (iPTH levels were positively correlated with baPWV, r = 0.27, *p* = 0.011) bone ALP = 12.5 μg/L (9.9–15.9)
[82]	N1: Ca = 9.1 mg/dL (8.7–9.4) vs. N2: Ca = 9.2 mg/dL (8.9–9.5)

Abbreviations: ACS = autonomous cortisol secretion; AI = adrenal incidentalomas, ALP = alkaline phosphatase; BAI = bilateral adrenal incidentaloma; BMD = bone mineral density; F = female; FN = femoral neck; LS = lumbar spine; M = male; MACS = mild autonomous cortisol secretion; N = number of patients; NFAI = non-functioning adrenal incidentaloma; OP = osteopenia; OR = odds ratio; OS = osteoporosis; SCS = subclinical Cushing syndrome; SD = standard deviation; UAI = unilateral adrenal incidentaloma, vs. = versus; CTX = serum C-telopeptide of type I collagen; D-Pyr/Cr = urinary deoxypyridinoline/creatinine; ICTP = carboxy-terminal cross-linked telopeptide of type I collagen; iPTH = intact parathyroid hormone; PICP = Carboxy-terminal propeptide of type I procollagen; PIIINP = amino-terminal propeptide of type III procollagen; PINP = N-terminal propeptide of type 1 collagen; PTH = parathyroid hormone; U-NTX = urinary N-terminal crosslinking telopeptide of type I collagen; 25OHD 25-hydroxyvitamin D.

**Table 6 jcm-12-04244-t006:** Longitudinal studies that followed bone status in patients with AIs (studies from 2009 to 2021) [57,62,79,80].

First Author Publication Year Reference Number Study Design Follow-Up Period	Baseline Status		Follow-Up	
Tauchmanova 2009 [57] Case-control study N = 46 N1 = 23 with SCS treated with clodronate N2 = 23 with SCS untreated **Follow-up: 1 y**	Treated-N1 LS BMD = 0.97 ± 0.12 g/cm^2^ FN BMD = 0.816 ± 0.14 g/cm^2^ Osteocalcin = 15 ± 3 ng/mL CTX = 1520 ± 923 pmol/L bone ALP = 14.2 ± 4 μg/L Ca = 2.35 ± 0.1 mmol/L iPTH = 43 ± 12 ng/L VFx = 65%	Untreated-N2 LS BMD = 0.98 ± 0.13 g/cm^2^ FN BMD = 0.817 ± 0.11 g/cm^2^ Osteocalcin = 14.6 ± 3.2 ng/mL CTX = 1590 ± 1065 pmol/L bone ALP = 15 ± 3.5 μg/L Ca = 2.34 ± 0.1 mmol/L iPTH = 38 ± 15 ng/L VFx = 61%	Treated-N1 LS BMD = 1.03 ± 0.11 g/cm^2^ FN BMD = 0.825 ± 0.1 g/cm^2^ Osteocalcin = 11.2 ± 2.8 ng/mL CTX = 1005 ± 670 pmol/L bone ALP = 11.1 ± 4.3 μg/L VFx = 65%	Untreated-N2 LS BMD = 0.96 ± 0.11 g/cm^2^ FN BMD = 0.8 ± 0.12 g/cm^2^ Osteocalcin = 13.8 ± 3.1 ng/mL CTX = 1533 ± 923 pmol/L bone ALP = 14.2 ± 4.2 μg/L VFx = 65%
Morelli 2011 [62] Cross-sectional study N = 103 women N1 = 27 with SH N2 = 76 without SH **Follow-up: 2 y**	With SH LS BMD (Z-score) = 0.01 ± 1.17 SD (−1.8–2.5) FN BMD (Z-score) = −0.04 ± 0.99 SD (−2.4–2.7) SDI = 1.11 ± 1.50(0–6) VFx = 55.6%	Without SH LS BMD (Z-score) = 0.03 ± 1.38 SD (−2.8–4.1) FN BMD (Z-score) = 0.07 ± 0.78 SD (−1.6–2.1) SDI = 0.58 ± 1.10 (0–4) VFx = 28.9%	With SH LS BMD (Z-score) = 0.27 ± 1.37 SD (−2.0–3.6) FN BMD (Z-score) = 0.00 ± 1.07 SD (−2.4–2.6) SDI = 2.11 ± 1.85 (0–8) VFx = 81.5% new VFx = 48.1%	Without SH LS BMD (Z-score) = 0.16 ± 1.45 SD (−2.6–4.6) FN BMD (Z-score) = 0.11 ± 0.83 SD (−1.7–2.9) SDI = 0.79 ± 1.40 (0–6) VFx = 35.5% new VFx = 13.2%
Podbregar 2021 [79] Prospective study N = 67 with NFAI **Follow-up: 10.5 y**	Osteoporosis prevalence = 17.9%		Osteoporosis prevalence = 26.9% (*p* = 0.031)	
Li 2021 [80] Cohort study N1 = 1004 with AI N2 = 1004 controls **Follow-up: 10 y**	N1 Any osteoporotic site = 16.6% VFx = 6.4% Hip fracture = 2.9% Distal forearm fracture = 9.4% N2 Any osteoporotic site = 13.3% (*p* = 0.04) VFx = 3.6% (*p* = 0.004) Hip fracture = 2% (*p* = 0.19) Distal forearm fracture = 9% (*p* = 0.76)		N1 Cumulative incidence of new VFx = 3.5% Cumulative incidence of new hip fracture = 2.6% Cumulative incidence of new distal forearm fracture = 3.2% N2 Cumulative incidence of new VFx = 3.6% (*p* = 0.33) Cumulative incidence of new hip fracture = 2.7% (*p* = 0.84) Cumulative incidence of new distal forearm fracture = 3% (*p* = 0.63)	

Abbreviations: ALP = alkaline phosphatase; BMD = bone mineral density; Ca = serum calcium; SCS = subclinical Cushing syndrome; CTX = serum C-telopeptide of type I collagen, FN = femoral neck; iPTH = intact parathyroid hormone; LS = lumbar spine; N = number of patients; SD = standard deviation; SDI = Spinal deformity index; SCS = subclinical Cushing syndrome; SH = subclinical hypercortisolism; VFx = vertebral fractures; y = years; 25OHD = 25-hydroxyvitamin D.

**Table 7 jcm-12-04244-t007:** Studies of patients diagnosed with AIs and ACS: the impact of adrenalectomy on bone status (starts with the oldest study) [60,64,68,70].

First Author, Publication Year Reference Number Study Design Studied Population	N1 (Patients with Adrenalectomy)		N2 (Conservative or Non-Surgical Group)	
	Before Adrenalectomy	After Adrenalectomy	At Baseline	At the End of Follow-Up
Toniato 2009 [60] Prospective study N = 45 N1 = 23 with adrenalectomy N2 = 22 with conservative	DXA (T-score) = −1.81 ± 2.07 SD 21.7% with osteoporosis	Any changes in bone parameters were detected after surgery	DXA(T-score) = −1.86 ± 1.93 SD 27.3% with osteoporosis	Median period of follow-up: 7.7 (2–17) years
Iacobone 2012 [64] Prospective study N = 35 N1 = 22 with adrenalectomy N2 = 15 with conservative	LS (T-score) = −1.23 ± 0.74 SD 30% with osteoporosis or osteopenia	Mean period of follow-up: 54 ± 34 months LS(T-score) = −1.29 ± 0.77 SD One “osteopenic” patient became “osteoporotic” (*p* = NS)	LS (T-score) = −1.19 ± 0.74 SD 26.7% with osteoporosis or osteopenia	Mean period of follow-up: 56 ± 37 months LS(T-score) = 1.27 ± 0.82 SD One “osteopenic” patient became “osteoporotic” (*p* = NS)
Perogamvros 2015 [68] Retrospective study N = 33 N1 = 14 with adrenalectomy N2 = 19 with conservative	3 patients with osteoporosis	Mean period of follow-up: 53.9 ± 21.3 months 2 patients with improvement of osteoporosis features	5 patients with osteoporosis	Mean period of follow-up: 51.8 ± 20.1 months no patient with improvement of osteoporosis features
Salcuni 2016 [70] Cross-sectional Prospective intervention N = 55 N1 = 32 with adrenalectomy N2 = 23 conservative	LS BMD (Z-score) = −0.86 ± 1.27 SD (−2.9–1.9) FN BMD (Z-score) = −0.54 ± 0.9 SD (−1.9–1.7) 25OHD = 21.9 ± 8.7 ng/mL Patients with VFx = 46.9%	LS BMD (Z-score) = −0.49 ± 1.17 SD (−2.7–2.1) LS ΔZ-score/year = 0.10 ± 0.2 SD (0.0–1.0) FN BMD (Z-score) = −0.41 ± 0.9 SD (−1.7–1.8) FN ΔZ-score/year = −0.09 ± 2.0 SD (−4.26–4.76) 25OHD = 39.4 ± 8.1 ng/mL Patients with VFx = 46.9% Patients with new VFx = 9.4%	LS BMD (Z-score) = 0.23 ± 1.4 SD (−1.8–2.7) FN BMD (Z-score) = 0.14 ± 1.2 SD (−2.4–2.7) 25OHD = 22.4 ± 7.3 ng/mL Patients with VFx = 65.2%	LS BMD (Z-score) = 0.25 ± 1.5 SD (−2.1–2.8) LS ΔZ-score/year = −0.01 ± 0.3 SD (0.03–0.17) FN BMD (Z-score) = 0.12 ± 1.2 SD (−2.4–2.6) FN ΔZ-score/year = −0.58 ± 2.3 SD (−83–0.64) 25OHD = 38.7 ± 7.5 ng/mL Patients with VFx = 91.3% Patients with new VFx = 52.2%

Abbreviations: BMD = bone mineral density; DXA = dual energy X-ray absorptiometry; FN = femoral neck; LS = lumbar spine; N = number of patients; SD = standard deviation; VFx = vertebral fractures; y = years; 25OHD 25-hydroxyvitamin D.

**Table 8 jcm-12-04244-t008:** Adrenal (cortisol excess assays) evaluations in patients diagnosed with osteoporosis and fractures [83,84].

First Author Publication Year Reference Number Study Design	Studied Population Gender (F/M) Age (Years)	Criteria for ACS Diagnosis	Prevalence of ACS	DXA–BMD, T-Score or Z-Score	Fracture Prevalence
Chiodini 2007 [83] Cross-sectional study	N = 219 N1 = 147 with osteoporosis F/M = 131/16 Mean age: 63.7 y N2 = 72 without osteoporosis F/M = 69/3 Mean age: 57.4 y	Serum cortisol after LDDST > 50.0 nmol/L UFC > 165.6 nmol/L and/or midnight cortisol > 207 nmol/L	Prevalence of SH: N1–4.8% (1.32%−8.20%) N2:0% (*p* = 0.099)	N1: Mean T-score (SD) LS = −2.88 (1.08) FT = −1.96 (0.88) FN = −2.03 (0.88) N2: Mean T-score (SD) LS = −1.37 (1.09) FT = −0.99 (0.97) FN = −1.13 (0.90)	N1: Grade of fracture (I/II/III), N = 54/14/8 N2: Grade of fracture (I/II/III), N = 0
Pugliese 2021 [84] Cross-sectional study	N = 101 patients with fractures N1 = 5 with less severe hypercortisolism (LSH) F/M = 3/2 Mean age: 62.6 ± 13.3 y N2 = 96 with non-LSH patients F/M = 72/24 Mean age: 65 ± 10.3 y	Serum cortisol after 1 mg DST > 1.8 μg/dL for confirmation Serum cortisol after LDDST > 1.8 μg/dL	Prevalence of LSH: 5.0%	N1: LS BMD (Z-score) = −1.5 ± 1.1 SD FN BMD (Z-score)= −1.2 ± 0.6 SD N2: LS BMD (Z-score) = −1.5 ± 1.3 SD FN BMD (Z-score) = −0.9 ± 0.9 SD *p* = NS	All

Abbreviations: BMD = bone mineral density; DST = dexamethasone suppression test; F = female; FN = femoral neck; FT = total femur; LDDST = low-dose dexamethasone suppression test; LS = lumbar spine; LSH = less severe hypercortisolism; M = male; N = number of patients; NS = not significant; SD = standard deviation; SH = subclinical hypercortisolism; UFC = Urinary free cortisol; y = years.

## Data Availability

Not applicable.

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
