# Peer review of "Management of Adrenal Cortical Adenomas: Assessment of Bone Status in Patients with (Non-Functioning) Adrenal Incidentalomas"

_jcm, 2023, doi:10.3390/jcm12134244_

Round 1
Reviewer 1 Report
Reviewer comments on manuscript JCM #2421716, entitled “Management of adrenal cortical adenomas: assessment of bone status in patients with (non-functioning) adrenal incidentalomas”,
Comment # 1, whole manuscript. The authors should carefully review the English writing, as in the present form the manuscript may distract the reader. Some (but not all) examples are pointed below:
- line 458 - “4 studied followed the pa-…”, please consider “Four” (instead of 4) after the stop and correct “studied” substituting for studies.
- line 539 - “We consider that the most difficult aspect of studding bone profile” please substitute for studying
- line 501 - “...respective 53.9±21.3 501 months in two cohorts (after adrenalectomy), respective a mean...”, please consider respectively instead of respective (which is wrong).
- lines 634-636 – “Overall, the prevalence of osteoporosis complicated or no with fragility fractures, especially at vertebral levels, is higher in persons with SH versus non-SH, for example, of 87.5% versus 27.8% (among the highest ratios), respectively of 68.8% versus 18.5% [66], of 75% versus 65%, respectively of 74% versus 55.6% [75].”
- line 653 – “Of note, adrenalectomy should be taken in consideration when decide anti-osteoporosis strategy”
- lines 662-664: “Also, the management (short and long term medical and surgical strategy) needs to integrate bone status to the larger panel of cardiovascular and metabolic complications due to ACS/SCS [108-110]; but, also, generally in patients with AIs/non-functioning AIs [111-117].” After de semicolon, the remaining phrase seems redundant. Please consider reformulate it.
- line 672: “In patients with ACS/SH, primary (menopausal or age-related osteoporosis) might overlap in addition to type 2 diabetes mellitus...”; this phrase is to confuse and will certainly distract the reader.
- line 699: “Anomalies of bone profile up to the diagnostic of osteoporosis and even complicated with fragility fractures represents an important element...”
Comment # 2, whole manuscript, including title: the authors perform a review on the bone impact of mild autonomous cortisol secretion (MACS). Thus, although clinically silent in terms of the classical signs of Cushing’s, the term “non-functioning” is problematic as the patients included have a non-suppressed cortisol after dexamethasone test and classical detrimental effects of MACS in terms of morbidity. Thus, the word in brackets “non-functioning” does not make too much sense. Please consider its reformulation/elimination. I will suggest the authors to consider MACS only (please be careful with the miswriting of this abbreviation – line 221).
Comment # 3, methods section: the authors performed two distinct analysis considering studies with different methodologies, namely prevalence of bone abnormalities in patients with MACS and prevalence of MACS in patients with osteoporosis. It should be clearly specified in the present manuscript methodology that these two types of studies are analyzed separately, as they include different populations with different characteristics (thus not suitable for pooled analysis). They should also insert in the body of the manuscript the reference numbers of the two studies that explore the prevalence of MACS in patients with osteoporosis.
Comment #4, line 548 (but not only): “...the values of the 2nd day (post 1 mg DST) ...”; 1 mg overnight dexamethasone suppression test has implicitly blood cortisol drawn in the next morning. This is well-known by who will read the manuscript and the authors can suppress “values of the 2nd day” (which may lead to some confusion).
The authors should carefully review the English writing, as in the present form the manuscript may distract the reader.
Author Response
Response to Review 1 Comments
Dear Reviewer,
Thank you very much for your time and your effort to review our manuscript.
We are very grateful for providing your valuable feedback on the article.
Here is our response and related amendment that has been made in the manuscript according to your review (marked in yellow color).
Reviewer comments on manuscript JCM #2421716, entitled “Management of adrenal cortical adenomas: assessment of bone status in patients with (non-functioning) adrenal incidentalomas”,
Comment # 1, whole manuscript. The authors should carefully review the English writing, as in the present form the manuscript may distract the reader.
Thank you. We reviewed the English writing.
Some (but not all) examples are pointed below:
- line 458 - “4 studied followed the pa-…”, please consider “Four” (instead of 4) after the stop and correct “studied” substituting for studies.
Thank you. We correct it.
- line 539 - “We consider that the most difficult aspect of studding bone profile” please substitute for studying
Thank you. We correct it.
- line 501 - “...respective 53.9±21.3 501 months in two cohorts (after adrenalectomy), respective a mean...”, please consider respectively instead of respective (which is wrong).
Thank you. We correct it.
- lines 634-636 – “Overall, the prevalence of osteoporosis complicated or no with fragility fractures, especially at vertebral levels, is higher in persons with SH versus non-SH, for example, of 87.5% versus 27.8% (among the highest ratios), respectively of 68.8% versus 18.5% [66], of 75% versus 65%, respectively of 74% versus 55.6% [75].”
Thank you. We correct it.
- line 653 – “Of note, adrenalectomy should be taken in consideration when decide anti-osteoporosis strategy”
Thank you. We correct it.
- lines 662-664: “Also, the management (short and long term medical and surgical strategy) needs to integrate bone status to the larger panel of cardiovascular and metabolic complications due to ACS/SCS [108-110]; but, also, generally in patients with AIs/non-functioning AIs [111-117].” After de semicolon, the remaining phrase seems redundant. Please consider reformulate it.
Thank you. We respectfully consider that the bone status should be particularly addressed in the subgroup of patients that displays mild (excessive) cortisol activity as defined by various criteria and terms over the years as we mentioned it (autonomous cortisol secretion or subclinical Cushing’s syndrome or subclinical hypercortisolism), but also, generally, in all the patients confirmed with adrenal incidentalomas, even at first they might seem not functional since these individuals might become functional (meaning to produce a small abnormal amount of cortisol) during follow-up or they be misdiagnosed as non-functional at first presentation due to various tests and cutoffs that have been used and then to be re-categorized as ACS/SCS. That is why in this sentence we highlight the importance of bone status assessment in particular (ACS subgroup), but, also, in general with respect to the mentioned tumors (AI). Thank you
- line 672: “In patients with ACS/SH, primary (menopausal or age-related osteoporosis) might overlap in addition to type 2 diabetes mellitus...”; this phrase is to confuse and will certainly distract the reader.
Thank you. We correct it.
- line 699: “Anomalies of bone profile up to the diagnostic of osteoporosis and even complicated with fragility fractures represents an important element...”
Thank you. We correct it.
Comment # 2, whole manuscript, including title: the authors perform a review on the bone impact of mild autonomous cortisol secretion (MACS). Thus, although clinically silent in terms of the classical signs of Cushing’s, the term “non-functioning” is problematic as the patients included have a non-suppressed cortisol after dexamethasone test and classical detrimental effects of MACS in terms of morbidity. Thus, the word in brackets “non-functioning” does not make too much sense. Please consider its reformulation/elimination. I will suggest the authors to consider MACS only (please be careful with the miswriting of this abbreviation – line 221).
Thank you very much. We respectfully need to mention (as within the manuscript) that the terminology does not belong to us. Over the years (meaning more than 3 decades) various terms have been used and currently there is no unanimous consent, either. In order to stress out the domain of the current analysis (involving bone and adrenal domains in relationship with adrenal adenomas of the cortex without typical clinical picture of hormonal excess, an approach which is a mostly innovative and challenging, this study on published data being one of the most complex of its kind) it is mandatory to use all the terms that even changed over time or are still used but not by everybody. We cannot change the terms applied by the authors of one paper published, for instance, at the moment when “subclinical Cushing’s syndrome” was frequently used which today is less encouraged to be defined as such. (Of note, subclinical Cushing’s syndrome does not mean that the patient is completely asymptomatic, but that he/she does not present the typical, full-blown manifested picture of well-known Cushing’s syndrome).
Also, recent papers use the term “autonomous cortisol secretion”, but in order to integrate older studies (before the year of 2016 where some important European societies in the matter of adrenal tumors released a position paper in order to encourage the new terminology) we need to mention all the terms. This is not up to us.
“MACS” term is less used, and we cannot exclusively mention it unless the authors of an original study used it such. Moreover, the defining criteria of autonomous cortisol secretion, subclinical Cushing’s syndrome, subclinical hypercortisolemia and mild autonomous cortisol secretion varied with the decade, study, or research group, as we mentioned it. This represents an additional challenge for our paper to cover all these data which in fact relate to the same well known sub-category of endocrine patients.
Additionally, another controversy relates to the definition of an adrenal incidentaloma which generally means an accidentally detected tumor via any radiological or imaging procedure. However, this is a general term, not a particular endocrine term. The addressability is multidisciplinary. But, in endocrinology, there is another specification, meaning a “non-functioning adrenal incidentaloma” that implies that not only a tumor was accidentally detected by preforming any type of adrenal glands scans such as ultrasound, computed tomography or magnetic resonance imaging, etc., but, also, a hormonal workup has been done (such as dexamethasone suppression test, metanephrines and/or normetanephrines assays or even screening tests for primary aldosteronism) and mostly found negative (with the exception of mild hormonal anomalies as seen in ACS/SCS). That is why we consider mandatory introducing the mentioned specification within the title. Moreover, some of the studies we analyzed observed anomalies of bone profile in the general category of non-functioning adrenal incidentaloma in relationship or no with the subgroup identified as ACS/SCS/SH which explains our approach of the topic. Thank you.
Comment # 3, methods section: the authors performed two distinct analysis considering studies with different methodologies, namely prevalence of bone abnormalities in patients with MACS and prevalence of MACS in patients with osteoporosis. It should be clearly specified in the present manuscript methodology that these two types of studies are analyzed separately, as they include different populations with different characteristics (thus not suitable for pooled analysis). They should also insert in the body of the manuscript the reference numbers of the two studies that explore the prevalence of MACS in patients with osteoporosis.
Thank you very much for this useful observation. We kept the original analysis at Results and mentioned the secondary analysis at Discussion which, indeed, targets a distinct population with different features (and it is not the main focus on this paper). Figure 1 introduces the flowchart of the first type of analysis which represents the focus of our sample – based study. Thank you for pointing this out. We really appreciate it. We correct it.
Comment #4, line 548 (but not only): “...the values of the 2nd day (post 1 mg DST) ...”; 1 mg overnight dexamethasone suppression test has implicitly blood cortisol drawn in the next morning. This is well-known by who will read the manuscript and the authors can suppress “values of the 2nd day” (which may lead to some confusion).
Thank you very much. We removed it.
The authors should carefully review the English writing, as in the present form the manuscript may distract the reader.
Thank you. We reviewed the English writing.
Thank you very much.
Reviewer 2 Report
The authors have presented a very ambitious review of bone status in patients with adrenal adenomas. It was written initially as a PhD thesis and is presented as a narrative review. The conclusions are that ACS has been variably defined over three decades, that studies are small, and bone status outcomes are heterogeneous.
The abstract and narrative are overly long which may prevent readers from reviewing the manuscript. The manuscript may benefit from editorial review. The term “Sine qua non” is used inappropriately both in the abstract and in the conclusion.
The issue for review is whether CS (equivalent) and ACS ( equivalent) are associated with bone pathology more than that found in AI (non functioning).
As written the narrative is very confusing. The tables are set up in a manner that precludes review. The manuscript needs extensive modification to improve clarity.
I recommend that table 1 should be revised into 6 columns headed by: study identifiers, CS (equivalent), ACS (equivalent), AI (nonfunctioning), Other controls, P values. Each row would have a parameter. If an article does not have comparison groups, then the article should be eliminated.
Table 1b should have the 3 columns, complete study identifiers as in Ia ( readers should not have to go back), column for criteria for CS, column for Criteria for ACS.
Table 2 should have 5 columns, Study identifiers, CS, ACS, AI, Other , P values. Each parameter should have a row.
Table 3 should have columns as above, since it should not be incumbent on the reader to go back to figure out which is N1 or N2.
Table 4 should have 5 columns as above table 2. Since this discussion is about the effects of endogenous steroids on bone, data for calcium 25OHD and iPTH are not relevant.
Table 5 should have 7 columns , study identifier, CS treated, CS untreated, p value, ACS treated ACS untreated, p value. Each row should have one parameter for comparison . Note here and narrative that clodronate is not currently used or available.
Table 6 should have 7 columns as table 6. Here again leave out data on 25OHD.
Table 7 and section 3.8 are not relevant to this review unless the primary article authors performed imaging studies and this table and section should be removed.
Each results section3.2-3.7 should start with a simple summary of the findings in the section. The authors then may go on with their narrative review that explains their summary.
Errors in syntax. May benefit from editorial review.
Author Response
Dear Editor,
Response to Review 2 Comments
Dear Reviewer,
Thank you very much for your time and your effort to review our manuscript.
We are very grateful for your insightful comments and observations, also, for providing your valuable feedback on the article.
Here is a point-by-point response and related amendments that have been made in the manuscript according to your review (marked in yellow color).
The authors have presented a very ambitious review of bone status in patients with adrenal adenomas.
Thank you very much. We really appreciate it.
It was written initially as a PhD thesis and is presented as a narrative review.
Thank you. The article is not at PhD thesis, neither had we mentioned it such. The article is part of the literature research that is mandatory in order to start an original study (including during PhD years). We mentioned the title of the PhD thesis at Acknowledgment section and its number contract at our University which is dated in October 2022. In any case, a PhD thesis cannot be written within a few months since initiating a PhD, neither it belongs to the actual PhD thesis itself which should be finished within 4 years. Also, due to the complexity of the article, this paper highlights a multidisciplinary collaboration beyond the endocrine issues, as we mentioned it. Thank you
The conclusions are that ACS has been variably defined over three decades, which studies are small, and bone status outcomes are heterogeneous.
Indeed, the level of statistical evidence, the changes of the terminology and cut off criteria, and the trans-disciplinary domains (bone-adrenal) represents the elements that raise the difficulty of this sample-based analysis which is one of the most complexes of its kind. Thank you.
The abstract and narrative are overly long which may prevent readers from reviewing the manuscript. The manuscript may benefit from editorial review.
Thank you. We reviewed the English writing. The Abstract is intended to capture the readers’ interest and it is allowed in the system in this format. The length of this type of paper is not limited at MDPI. Neither such complex analysis could be reduced since we covered all aspects of bone assessments in this particular type of tumors for the last 3 decades. Thank you.
The term “Sine qua non” is used inappropriately both in the abstract and in the conclusion.
Thank you. We replaced it.
The issue for review is whether CS (equivalent) and ACS (equivalent) are associated with bone pathology more than that found in AI (non functioning). As written the narrative is very confusing. The tables are set up in a manner that precludes review. The manuscript needs extensive modification to improve clarity.
Thank you. We did not include CS (Cushing’s syndrome) unless the study compared it with a subgroup of analysis regarding patients confirmed with ACS/SCS/SH (which we already mentioned it). The terms of ACS, SCS, SH or MACS do not belong to us. They have been designated over the years for the same sub-group of patients from the larger group of adrenal incidentalomas/tumors. Some authors studied the specific subgroups and some addressed individuals with adrenal incidentalomas (a part of them display mild cortisol excess designated as ACS) that might have bone anomalies due to the same cortisol issues, but not studied as such. The fact that terminology varied over the decades, as well as the cutoffs, criteria and subgroups of analysis, studied population according to each study design and assessments is captured within our study that makes it complex. It cannot be a unifying manner since it is mandatory to take into consideration the designation of each original endocrine study and presented as such (according to original authors) – as cited at references [43-82]. Thank you.
I recommend that table 1 should be revised into 6 columns headed by: study identifiers, CS (equivalent), ACS (equivalent), AI (nonfunctioning), Other controls, P values. Each row would have a parameter. If an article does not have comparison groups, then the article should be eliminated.
Thank you. We respectfully mention the followings: Table 1A. We did not include CS (equivalent) unless the study compare it to a subgroup of subclinical or preclinical CS (or ACS, etc.) – references 43, 45, 46, 51, 56, 72, 76. We introduced green color for CS, blue color for AI/NFAI and red color for ACS (equivalent) and mentioned it such. This is a population description; no p-values are included. There are no other parameters than the definition of ACS/AI as it was applied in each study since their design varies a lot. We did not include as exclusion criteria studies without a comparison group. This is a narrative, not a systematic review and it is up to the authors the inclusion criteria. We particularly choose this type of paper not to limit the enrolled population in published data since a great variation of study design and studied population is found over the last 30 years. Thank you
Table 1b should have the 3 columns, complete study identifiers as in Ia (readers should not have to go back), column for criteria for CS, column for Criteria for ACS.
Thank you. We changed the first column according to your suggestion. The second column includes the focus of our current study – the definition of ACS/SCS according to each study. We respectfully mention that our study does not include CS which is only additionally mentioned in a few cited studies and it does not represent the objective of this paper. Thank you.
Table 2 should have 5 columns, Study identifiers, CS, ACS, AI, Other , P values. Each parameter should have a row.
Thank you. According to your suggestion, each parameter has a row. P-value is already included (if provided by the original study) to the row of these parameters. We respectfully mention that no specific CS analysis is presented in this paper. At Methods we already introduced the followings: “Exclusion criteria: studies specially addressing clinically manifested (overt) CS….”
Table 3 should have columns as above, since it should not be incumbent on the reader to go back to figure out which is N1 or N2.
Thank you. We respectfully consider that we cannot repeat the same information for 3 times since it is already provided within the paper within the first two tables. Thank you for your understanding.
Table 4 should have 5 columns as above table 2. Since this discussion is about the effects of endogenous steroids on bone, data for calcium 25OHD and iPTH are not relevant.
Thank you. We corrected it according to your suggestion, but respectfully choose to keep all the information on mineral metabolism (including vitamin D and parathyroid hormone status) since they are a classical and mandatory part of (endocrine) bone assessment, a traditional evaluation of bone status in endocrinology (including with respect to the adrenal tumors). The same observations as above concerning the fact that we cannot repeat the studied population since it is already mentioned twice. According to your suggestions, each studied parameter has its row and associated p-values (if any). Thank you.
Table 5 should have 7 columns, study identifier, CS treated, CS untreated, p value, ACS treated ACS untreated, p value. Each row should have one parameter for comparison. Note here and narrative that clodronate is not currently used or available.
Thank you. We respectfully mention that in Table 5 there is no patient with CS. Only one study include data concerning specific therapy – reference no 57. We appreciate your recommendation to add to the main text the data on clodronate and we followed it: “Currently, the drug is no longer used or available in daily practice.“ Thank you.
Table 6 should have 7 columns as table 6. Here again leave out data on 25OHD.
Thank you. We respectfully mention above our response. Thank you
Table 7 and section 3.8 are not relevant to this review unless the primary article authors performed imaging studies and this table and section should be removed.
Thank you. In endocrinology, the decision of adrenalectomy for this type of tumors also takes into consideration the bone status, for example, if one patient continues to decrease bone mineral density without any other cause, the decision of adrenalectomy should be done since the cause might the tumor – related cortisol (even mild, but persistent over the years). Since this issue is still open and we could only found four such studies, we consider them very important in order to point out the fact that, after tumor removal, bone status might improve since the source of (even mild) autonomous cortisol excess has been removed from the body. Thank you
Each results section 3.2-3.7 should start with a simple summary of the findings in the section. The authors then may go on with their narrative review that explains their summary.
Thank you very much for your recommendation which we followed accordingly.
Chapter 3.2. DXA assessment in patients with AI (or non-functioning AI) provided the prevalence of osteoporosis/ostepenia or the BMD reports versus controls, time-dependent BMD changes or post-adrenalectomy bone effects including at DXA exam. As mentioned, ACS (or SCS or SH) involves a distinct type of tumour with positive (yet mild, not overt) cortisol (persistent) excess and, consecutively, DXA-BMD evaluation depends on the criteria of defining this hormonal activity, on the age and sex of studied population (including the menopausal status in females) and on the sample size of the cohort. Except for three, 37 cohorts provided data in terms of DXA and/or VFs [43,47-82] and generally a confirmation of negative impact of bone status is confirmed.
Chapter 3.3. Mild cortisol overproduction in patients with ACS (or SCS) might impair the skeleton qualitative features, as reflected by the bone microarchitecture analysis. Overall, we identified 5 such studies, particularly addressing TBS, but, also, spinal deformity index (SDI), and high-resolution peripheral quantitative computed tomography (HR-pQCT) [63,72,73,81]. The potential cortisol over-production from the tumour correlates with a negative impact at the level of micro-architecture while the level of statistical evidence is less convincing that seen in DXA-BMD analysis.
Chapter 3.4. BTMs represent additional tools in skeleton status assessments as a close reflection of persistent hypercortisolism and associated skeleton and mineral metabolism changes. Cortisol overproduction (even mild) might impair bone formation, but overall data are based on small sample size studies (of less than 100 patients per paper) [44-46,48-57,63,65,70,75,76,78,81,82].
Chapter 3.5. Approximately, one out of 10 cases with AI has a bilateral tumor, but a direct relationship with a more damaged bone profile still represents an open issue, mostly due to the lack of large cohorts specifically referring to bilateral opposite to unilateral adenomas with respect to skeleton status. We found 2 studies on bilateral lesions.
Chapter 3.6. Several studies addressed the issue of prevalent VFx (rarely, all types of fragility fractures, all referring to non-vertebral). (Table 1 and 2) Additional factors as hypogonadism (such as menopause) might contribute to VFx, although not all studied agree [52,55,59].
Chapter 3.7. The optimal management of patients with ACS associated a dynamic approach over the years due to continuous changing of definition criteria and specific indications which vary from an individual approach to guidelines recommendations. The importance of the topic is related to the fact that close follow-up of medical treatment for associated morbidities is needed, especially if surgical treatment is not chosen. Four studies followed the patients from 1 year to more than a decade [57,62,79,80].
Errors in syntax. May benefit from editorial review.
Thank you. We reviewed the English writing.
Thank you very much,
Round 2
Reviewer 2 Report
The authors were given recommendations to reorganize the data in the tables to make the manuscript more understandable. They did not.
The article may serve as a literature reference for others to analyze the effect of ACS.
Minor
Line 37. "Correlation" means there is a statistical r value. Either give the correlation coefficient here or in the text or change the word.
Typos line 58, 61
Contrary to the authors' opinion, the English syntax should have editorial assistance.
Author Response
Response to Review 2 Comments (Round 2)
Dear Reviewer,
Thank you very much for your time and your effort to review our manuscript for the second time.
We are very grateful for providing your valuable feedback on the article and for all the observations, suggestions and recommendations.
Here is our response and related amendment that has been made in the manuscript according to your review (marked in yellow color) in addition to prior revision.
The authors were given recommendations to reorganize the data in the tables to make the manuscript more understandable. They did not.
Thank you very much. We respectfully mention that we adjusted the tables as we prior explained. Thank you for your understanding.
The article may serve as a literature reference for others to analyze the effect of ACS.
Thank you very much. We really appreciate it!
Minor. Line 37. "Correlation" means there is a statistical r value. Either give the correlation coefficient here or in the text or change the word.
Thank you. We correct it.
Typos line 58, 61.
Thank you. We correct them.
Contrary to the authors' opinion, the English syntax should have editorial assistance.
Thank you. We checked again the English syntax. Thank you
Thank you very much.
